# Encapsulated Co–Ni alloy boosts high-temperature CO$_2$ electroreduction

Wenchao Ma[1], Jordi Morales-Vidal[2], Jiaming Tian[1], Meng-Ting Liu[3], Seongmin Jin[4], Wenhao Ren[1], Julian Taubmann[5], Christodoulos Chatzichristodoulou[5], Jeremy Luterbacher[4], Hao Ming Chen[3], Núria López[2] & Xile Hu[1]✉

Electrochemical CO$_2$ reduction into chemicals and fuels holds great promise for renewable energy storage and carbon recycling[1–3]. Although high-temperature CO$_2$ electroreduction in solid oxide electrolysis cells is industrially relevant, current catalysts have modest energy efficiency and a limited lifetime at high current densities, generally below 70% and 200 h, respectively, at 1 A cm$^{-2}$ and temperatures of 800 °C or higher[4–8]. Here we develop an encapsulated Co–Ni alloy catalyst using Sm$_2$O$_3$-doped CeO$_2$ that exhibits an energy efficiency of 90% and a lifetime of more than 2,000 h at 1 A cm$^{-2}$ for high-temperature CO$_2$-to-CO conversion at 800 °C. Its selectivity towards CO is about 100%, and its single-pass yield reaches 90%. We show that the efficacy of our catalyst arises from its unique encapsulated structure and optimized alloy composition, which simultaneously enable enhanced CO$_2$ adsorption, moderate CO adsorption and suppressed metal agglomeration. This work provides an efficient strategy for the design of catalysts for high-temperature reactions that overcomes the typical trade-off between activity and stability and has potential industrial applications.

Electrocatalytic CO$_2$ reduction to produce chemicals and fuels is a potentially important pathway towards a net-zero-emission society[1–3]. Extensive research has been conducted on low-temperature CO$_2$ electroreduction (below 100 °C), but such technology faces many challenges, including low system-level energy efficiencies and limited catalyst lifetimes[1,3]. The energy efficiency and lifetime of industrially relevant membrane electrode assembly (MEA) electrolysers used for low-temperature CO$_2$ electroreduction are typically less than 35% and 100 h, respectively, at current densities of 1 A cm$^{-2}$ or higher[1,9–11]. Furthermore, there is the intrinsic problem of carbonate formation resulting from the reaction between CO$_2$ and OH$^-$, which reduces the carbon efficiency and lifetime to unpractically low values[12]. In view of these problems, high-temperature CO$_2$ electroreduction (600–1,000 °C) in solid oxide electrolysis cells (SOEC) has emerged as an attractive approach for CO$_2$ utilization[4–8]. This approach uses pure CO$_2$ as the only reactant, without inclusion of H$_2$O or other hydrogen sources, thereby affording complete selectivity for CO formation[4,7] (Fig. 1a). Moreover, an energy efficiency of greater than 50% is achievable at 1 A cm$^{-2}$ for high-temperature CO$_2$ electroreduction[4,7].

Current CO$_2$ SOEC catalysts, which consist of pure metals[13], simple mixtures of metals and oxides[14–16], or composites of oxide support and metal decoration[17–19], either have limited numbers of active interfaces or are affected by severe particle agglomeration or coke formation at evaluated temperatures, leading to activity/stability trade-offs (Fig. 1b). Several strategies, including use of bimetallic catalysis[13,14], exsolution[17,18], redox cycling[19] and morphology engineering[20], have been explored to improve the performance of these catalysts. However, despite these efforts, the energy efficiencies and lifetimes achieved

with catalysts based on non-precious metals at industrially relevant current densities (1 A cm$^{-2}$ or greater) remain modest, typically below 70% and 200 h, respectively[14,16,19,21]. Although a lifetime of 1,000 h has been reported for a precious Ru–Fe alloy catalyst, this was achieved at a low current density of approximately 0.5 A cm$^{-2}$, and it degraded by about 60% thereafter[17].

Alloy engineering provides a strategy to modulate the surface electronic and geometric structure of metals, thereby enhancing their catalytic performances[17,22,23]. We propose that encapsulating active alloys within inert oxides could effectively prevent alloy agglomeration while creating rich interfaces, breaking the activity/stability trade-off (Fig. 1b). In this regard, we have designed a non-precious Co–Ni alloy catalyst encapsulated with Sm$_2$O$_3$-doped CeO$_2$ (SDC), which achieves both high activity and stability for high-temperature CO$_2$-to-CO conversion. The energy efficiency and lifetime of this catalyst reach 90% and 2,000 h at 1 A cm$^{-2}$ and 800 °C, respectively, surpassing those of state-of-the-art catalysts under similar conditions. The unique combination of encapsulation as the structural feature and efficient Co–Ni alloys contributes to enhanced triple-phase interfaces and suppresses surface reconstruction and coke formation, thereby enabling efficient and stable high-temperature CO$_2$ electroreduction.

## CO$_2$ electroreduction performance

M$_x$Ni$_{1-x}$ catalysts with a M/Ni molar ratio of $x$:(1 − $x$) encapsulated by SDC (denoted M$_x$Ni$_{1-x}$@SDC), where M represents a second non-precious metal and $x$ ranges from 0 to 1, were synthesized using a sol–gel method. Nickel was chosen as the host metal owing to its current efficiency as a

[1]Laboratory of Inorganic Synthesis and Catalysis, Institute of Chemical Sciences and Engineering, École Polytechnique Fédérale de Lausanne (EPFL), Lausanne, Switzerland. [2]Institute of Chemical Research of Catalonia (ICIQ-CERCA), The Barcelona Institute of Science and Technology, Tarragona, Spain. [3]Department of Chemistry and Center for Emerging Materials and Advanced Devices, National Taiwan University, Taipei, Taiwan. [4]Laboratory of Sustainable and Catalytic Processing, Institute of Chemical Sciences and Engineering, École Polytechnique Fédérale de Lausanne (EPFL), Lausanne, Switzerland. [5]Department of Energy Conversion and Storage, Technical University of Denmark, Kongens Lyngby, Denmark. ✉e-mail: xile.hu@epfl.ch

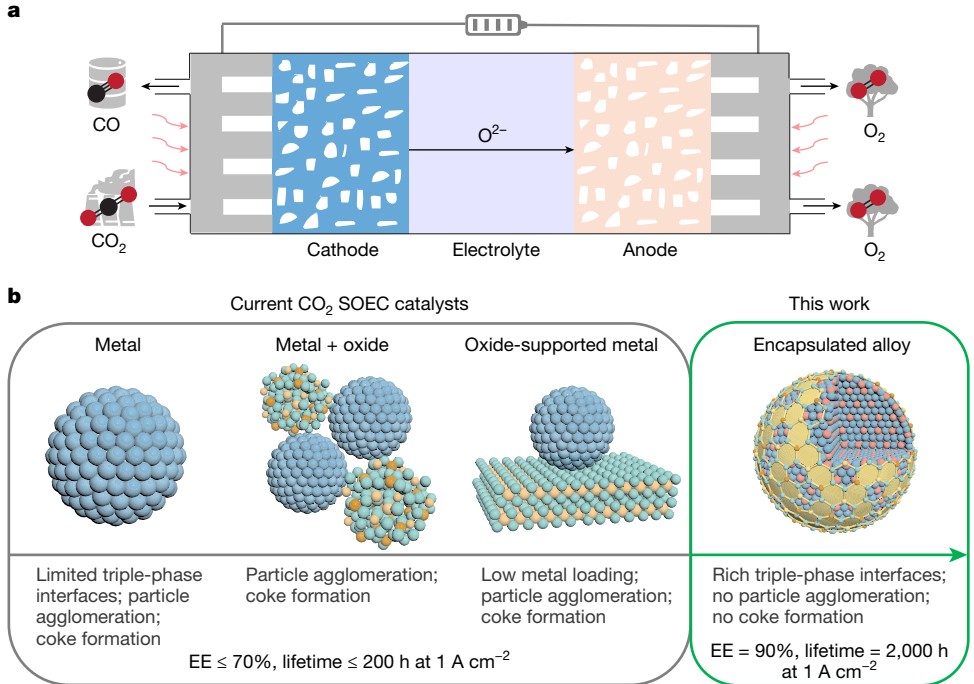

**Fig. 1 | Schematic illustration of high-temperature CO₂ electroreduction. a**, SOEC configuration. **b**, Overview of cathode catalysts developed for CO₂ SOEC.

non-precious metal catalyst for high-temperature CO₂ electroreduction[4,8], whereas SDC was selected as the oxide material owing to its high oxygen ion conductivity[24–26]. To evaluate the performance of the catalysts, we incorporated them into electrolyte-supported cells that consisted of a $M_xNi_{1-x}$@SDC cathode, an $La_{0.8}Sr_{0.2}Ga_{0.8}Mg_{0.2}O_{3-\delta}$ (LSGM) electrolyte, an $La_{0.6}Sr_{0.4}Co_{0.2}Fe_{0.8}O_{3-\delta}$ (LSCF) anode, and two SDC buffer layers to prevent side reactions between the electrode and electrolyte[4] (Supplementary Fig. 1 and Supplementary Table 1). In a preliminary screening of various second non-precious metals, the $Co_xNi_{1-x}$@SDC catalyst showed the highest current densities for CO₂ electroreduction to CO at all investigated cell voltages at 800 °C, while maintaining a Faradaic efficiency of CO ($FE_{CO}$) close to 100% (Supplementary Fig. 2). Correlation analysis of the Co and Ni molar ratios in the $Co_xNi_{1-x}$@SDC catalysts showed that the current density increased with increasing Co content up to $x$ = 0.5, and further increase in Co content instead decreased the current density (Supplementary Fig. 3). Consequently, $Co_{0.5}Ni_{0.5}$@SDC was selected for use in subsequent experiments.

We further synthesized two reference samples: a composite catalyst of $Co_{0.5}Ni_{0.5}$ and SDC without an encapsulated structure, referred to as $Co_{0.5}Ni_{0.5}$–SDC (with a composition similar to $Co_{0.5}Ni_{0.5}$@SDC, as shown in Supplementary Table 2); and an encapsulated Ni sole catalyst (Ni@SDC). For high-temperature CO₂ electroreduction, the current densities followed the sequence SDC < Ni@SDC < $Co_{0.5}Ni_{0.5}$–SDC < $Co_{0.5}Ni_{0.5}$@SDC at all investigated cell voltages (Fig. 2a), and the $FE_{CO}$ remained around 100% (Supplementary Fig. 4a). Notably, the current density over $Co_{0.5}Ni_{0.5}$@SDC reached 1.0 A cm⁻² at a cell voltage of only 1.1 V, approximately 1.5, 1.7 and 16.7 times higher than those over the $Co_{0.5}Ni_{0.5}$–SDC, Ni@SDC and SDC catalysts, respectively (Fig. 2a). We further normalized the current densities on the basis of electrochemical active surface areas (ECSA), and the normalized activity followed a consistent trend (Supplementary Figs. 4b and 5 and Supplementary Table 3). In addition, a physical mixture of Ni@SDC and Co@SDC with segregated Ni and Co phases (denoted Ni@SDC+Co@SDC) showed inferior activity compared with $Co_{0.5}Ni_{0.5}$@SDC featuring a Co–Ni alloy phase (Supplementary Figs. 6 and 7). These results demonstrate that both the alloy composition and the encapsulated structure have crucial roles in the high CO₂ electroreduction activity of $Co_{0.5}Ni_{0.5}$@SDC.

We further evaluated the energy efficiency and single-pass yield for CO₂-to-CO conversion in our system to demonstrate its practical applicability. The energy efficiency followed a similar sequence of SDC < Ni@SDC < $Co_{0.5}Ni_{0.5}$–SDC < $Co_{0.5}Ni_{0.5}$@SDC across all examined current densities and reached 90% or greater at current densities up to 1.0 A cm⁻² over $Co_{0.5}Ni_{0.5}$@SDC (Fig. 2b). A further increase in current density over $Co_{0.5}Ni_{0.5}$@SDC decreased energy efficiency owing to an increase in cell voltage, but energy efficiencies remained at 75% or greater at current densities up to 2.0 A cm⁻² (Fig. 2b). The single-pass yield of CO over $Co_{0.5}Ni_{0.5}$@SDC increased from about 22% to 90% as the flow rate of CO₂ decreased from 50 to 12 ml min⁻¹ at 1 A cm⁻², while a cell voltage of approximately 1.1 V was maintained (Fig. 2c). The CO selectivity on a molar carbon basis remained consistently close to 100% (Fig. 2c).

Stability is a critical metric for assessment of the performance of CO₂ electroreduction, particularly in high-rate systems with industrial relevance. To evaluate the stability of our catalysts, we conducted tests under a constant current density of 1.0 A cm⁻². The $Co_{0.5}Ni_{0.5}$@SDC catalyst with an encapsulated structure demonstrated exceptional long-term stability, as evidenced by a minor increase in cell voltage from 1.10 to 1.20 V after 2,000 h of operation, accompanied by an $FE_{CO}$ that was consistently close to 100% throughout the operation (Fig. 2d). The degradation rate for this catalyst was a mere 0.050 mV h⁻¹. By contrast, the $Co_{0.5}Ni_{0.5}$–SDC catalyst lacking an encapsulated structure, despite exhibiting high initial activity and approximately 100% $FE_{CO}$, showed a significant increase in cell voltage from 1.21 to 1.60 V after 800 h of operation (Fig. 2d), with a degradation rate of 0.49 mV h⁻¹. Similarly, the Ni@SDC catalyst demonstrated limited long-term stability, with the cell voltage rising from 1.33 to 1.60 V after 500 h of operation (Fig. 2d), corresponding to a degradation rate of 0.54 mV h⁻¹. In addition, the $FE_{CO}$ gradually decreased over time on the Ni@SDC catalyst (Fig. 2d), probably owing to coke formation[26,27]. The other $Co_xNi_{1-x}$@SDC catalysts ($x$ = 0.2, 0.75 and 1.0) also showed inferior stability compared with $Co_{0.5}Ni_{0.5}$@SDC (Supplementary Fig. 8). These findings suggest that both the encapsulated structure and the Co–Ni alloy composition contribute to the robustness of our $Co_{0.5}Ni_{0.5}$@SDC catalyst for high-temperature CO₂ electroreduction.

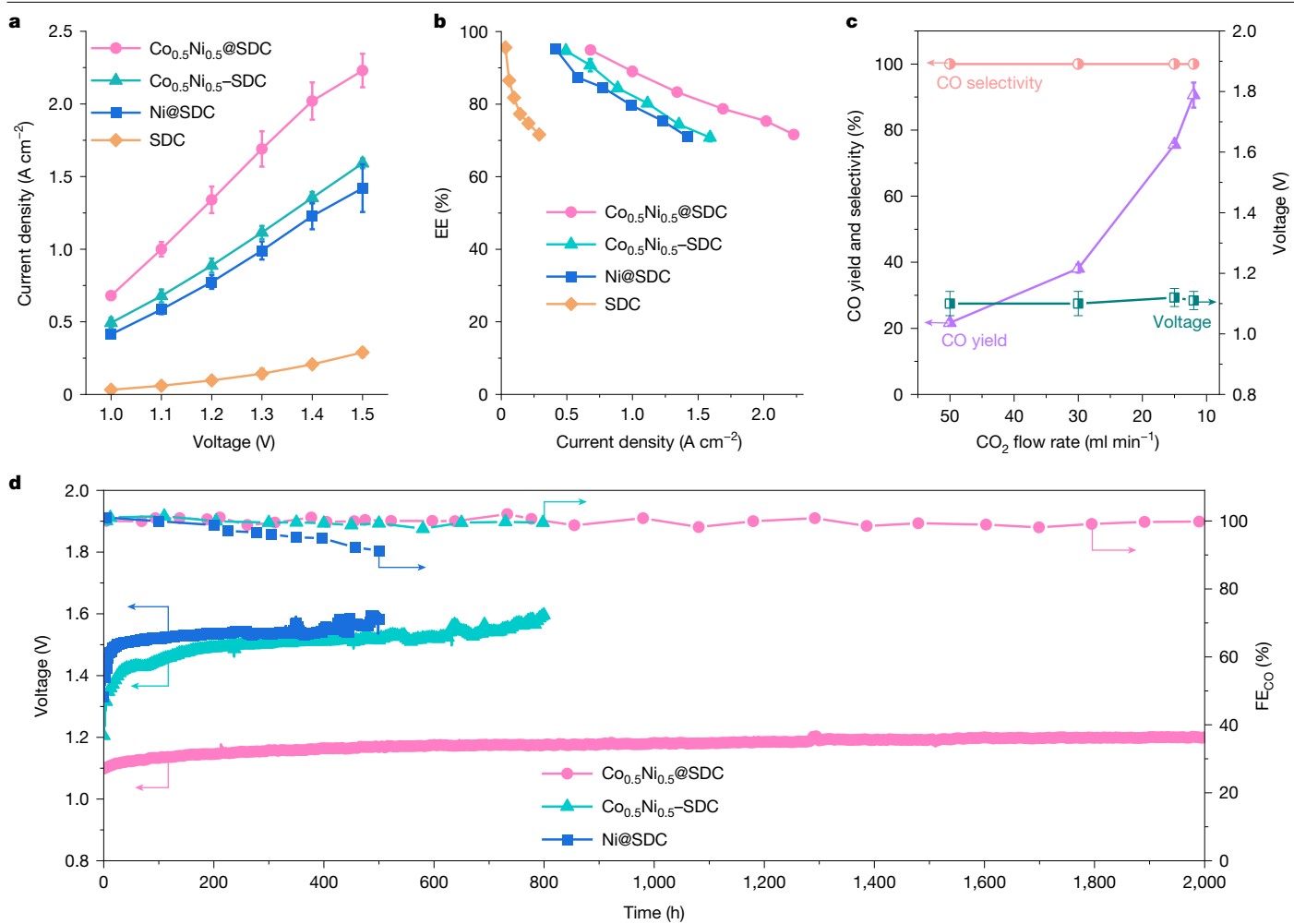

**Fig. 2 | CO₂ electroreduction performance. a**, Current densities at different cell voltages for different catalysts. **b**, Energy efficiencies at different current densities for different catalysts. **c**, Effects of CO₂ flow rate on CO single-pass yield and selectivity at 1 A cm⁻² over Co₀.₅Ni₀.₅@SDC. **d**, Stability tests at a constant current density of 1.0 A cm⁻². The results are shown as the mean ± s.d. from three individual experiments.

## Catalyst characterizations

We carried out various characterizations to confirm the composition and structure of the Co₀.₅Ni₀.₅@SDC (Fig. 3a), Co₀.₅Ni₀.₅–SDC (Fig. 3b) and Ni@SDC (Fig. 3c) catalysts. X-ray diffraction measurements showed that all catalysts consisted of face-centred cubic metals and fluorite SDC (Supplementary Fig. 6). The diffraction peaks corresponding to $Co_xNi_{1-x}$ were located between those of pure Ni and Co metals and underwent a negative shift with an increase in Co content (Supplementary Fig. 6), which could be attributed to the larger atomic radius of Co compared with that of Ni. This indicates mutual alloying of Co and Ni in our catalysts[28]. Energy-dispersive X-ray spectroscopy mappings using scanning electron microscopy and transmission electron microscopy (TEM) showed that all $Co_xNi_{1-x}$@SDC had an encapsulated structure: Co and/or Ni homogeneously formed a core, with the surface covered by Sm, Ce and O (Fig. 3d,f and Supplementary Figs. 9–11). By contrast, Co₀.₅Ni₀.₅–SDC had a random distribution of Co–Ni and Sm, Ce and O without an encapsulated morphology (Fig. 3e and Supplementary Figs. 9b and 10b). The homogeneous distribution of Co and Ni in Co₀.₅Ni₀.₅@SDC and Co₀.₅Ni₀.₅–SDC further confirmed the alloying of Co and Ni in both samples. Notably, the voids between the outer SDC particles could ensure easy access of CO₂ to the inner active interfaces over the $Co_xNi_{1-x}$@SDC catalysts (Fig. 3a,c). X-ray photoelectron spectroscopy (XPS) measurements revealed surface Co and Ni contents of approximately 15 mol.% for Co₀.₅Ni₀.₅@SDC and Ni@SDC,

lower than their respective bulk contents of approximately 80 mol.% as determined by inductively coupled plasma mass spectrometry (Supplementary Fig. 12). By contrast, the surface Co and Ni content for Co₀.₅Ni₀.₅–SDC was about 60 mol.%, close to the bulk content of approximately 80 mol.% (Supplementary Fig. 12). This result further confirmed the encapsulated structures of Co₀.₅Ni₀.₅@SDC and Ni@SDC and the composite structure of Co₀.₅Ni₀.₅–SDC.

The quasi in situ XPS spectra (Supplementary Fig. 13) of Ni 2p and Co 2p showed reduced binding energies of metallic Ni in Ni@SDC and Co in Co@SDC compared with their respective pure forms. The Ce 3d XPS spectra also showed decreased $Ce^{3+}$ contents in both samples compared with pure SDC. These trends suggest a charge transfer from $Ce^{3+}$ to either Ni or Co in each sample. This phenomenon persisted upon alloying, with the oxidation states of Co and Ni remaining below zero and a slight increase in $Ce^{3+}$ content compared with their monometallic counterparts. Such alterations are conducive to enhanced oxygen ion transport[4]. Moreover, Co₀.₅Ni₀.₅@SDC showed enhanced charge transfer between Co–Ni and SDC compared with Co₀.₅Ni₀.₅–SDC, suggesting an increased area of metal–oxide interfaces as a result of encapsulation. Consistent with the XPS findings, X-ray absorption near-edge structure (XANES) (Fig. 3g,h and Supplementary Table 4) and operando Raman spectra (Supplementary Fig. 14) corroborated the observed charge transfer from oxide to metal in all samples, with Co₀.₅Ni₀.₅@SDC demonstrating more metal–oxide interfaces than Co₀.₅Ni₀.₅–SDC. Extended X-ray absorption fine structure spectra (Supplementary Fig. 15 and

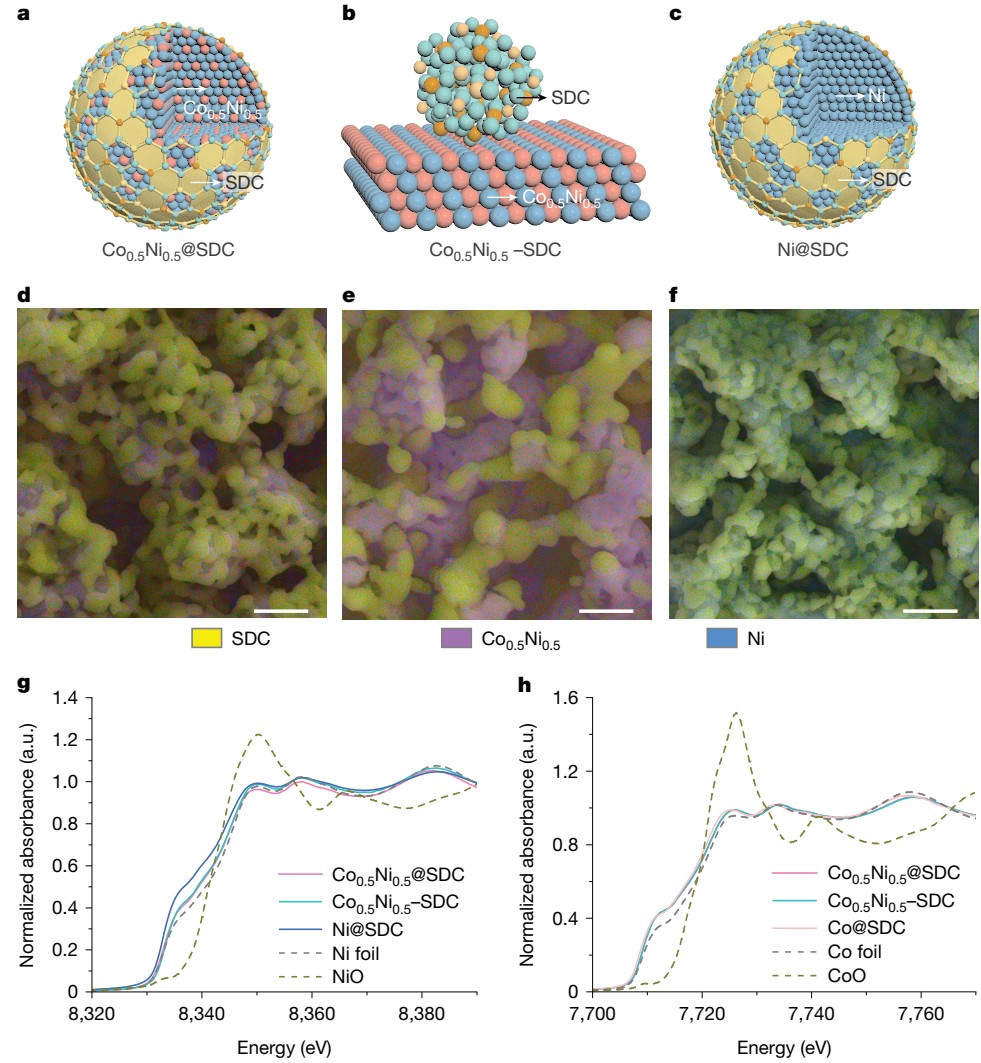

**Fig. 3 | Characterizations. a–c**, Schematic illustrations of the $Co_{0.5}Ni_{0.5}$@SDC (**a**), $Co_{0.5}Ni_{0.5}$–SDC (**b**) and Ni@SDC (**c**) catalysts. **d–f**, Merged scanning electron microscopy and energy-dispersive X-ray spectroscopy mappings for $Co_{0.5}Ni_{0.5}$@SDC (**d**), $Co_{0.5}Ni_{0.5}$–SDC (**e**) and Ni@SDC (**f**) catalysts. See Supplementary Fig. 9 for detailed distributions of each element. **g**, Ni K-edge XANES spectra. **h**, Co K-edge XANES spectra. Scale bars, 1 µm.

Supplementary Tables 5 and 6) showed similar bond lengths and slight variations in coordination numbers for Ni(Co)–Co(Ni) among the samples, probably owing to differences in particle sizes.

Postcatalytic characterizations showed that although there were no noticeable changes in the elemental composition and crystalline phase of any of the catalysts after the stability tests (Supplementary Fig. 16 and Supplementary Table 7), their morphologies showed distinct trends (Supplementary Figs. 17–20). The average particle sizes of metals in Ni@SDC and $Co_{0.5}Ni_{0.5}$–SDC increased notably by 60% and 100% following stability tests, respectively, whereas the increases in other $Co_xNi_{1-x}$@SDC catalysts (*x* = 0.2, 0.5, 0.75 and 1.0) featuring encapsulated structures were all below 15% (Supplementary Fig. 19). Meanwhile, the increases in average particle sizes of SDC in all catalysts remained below 10% (Supplementary Fig. 20). Raman measurements and carbon elemental analysis revealed coke formation on Ni@SDC after stability tests[26,27], whereas no carbon was detected in other catalysts (Supplementary Fig. 21 and Supplementary Table 8). Poststability XPS and XANES spectra revealed severe Co oxidation in Co@SDC, in contrast to the modest changes in the oxidation states of Co and Ni in Ni-containing catalysts (Supplementary Figs. 22 and 23); this was probably due to the higher oxygen affinity of Co relative to Ni[29]. Moreover, catalysts with higher Co content underwent mass loss (Supplementary Table 9). These results demonstrate that the encapsulated structure can

mitigate metal agglomeration, while the optimal Co–Ni alloy can suppress coke formation, metal oxidation and mass loss, thereby contributing synergistically to enhanced stability. The encapsulated structure in Ni@SDC failed to inhibit Ni agglomeration, possibly owing to coke formation on the Ni surface disrupting the encapsulated structure.

## Functioning mechanisms

To understand the variations in catalyst performance, we conducted operando electrochemical impedance spectroscopy (EIS) measurements and distribution of relaxation time (DRT) analyses under $CO_2$ and CO atmospheres. The EIS results (Fig. 4a,b and Supplementary Fig. 24) showed that with increasing Co content in $Co_xNi_{1-x}$@SDC, the polarization resistances ($R_p$) of the fuel electrode under $CO_2$ first decreased (up to *x* = 0.5) and then increased, whereas they continuously increased under CO. Compared with $Co_{0.5}Ni_{0.5}$–SDC, $Co_{0.5}Ni_{0.5}$@SDC exhibited lower $R_p$ values for the fuel electrode under both $CO_2$ and CO. All ohmic resistance ($R_o$), $R_p$ of the air electrode and double-layer capacitance values remained similar. In the DRT analysis, the high-frequency, intermediate-frequency and low-frequency peaks corresponded to oxygen ion migration in the electrolyte, surface oxygen transfer on the air side, and electrochemical adsorption and activation processes on the fuel side, respectively[17,30,31]. Increasing Co content in $Co_xNi_{1-x}$@SDC

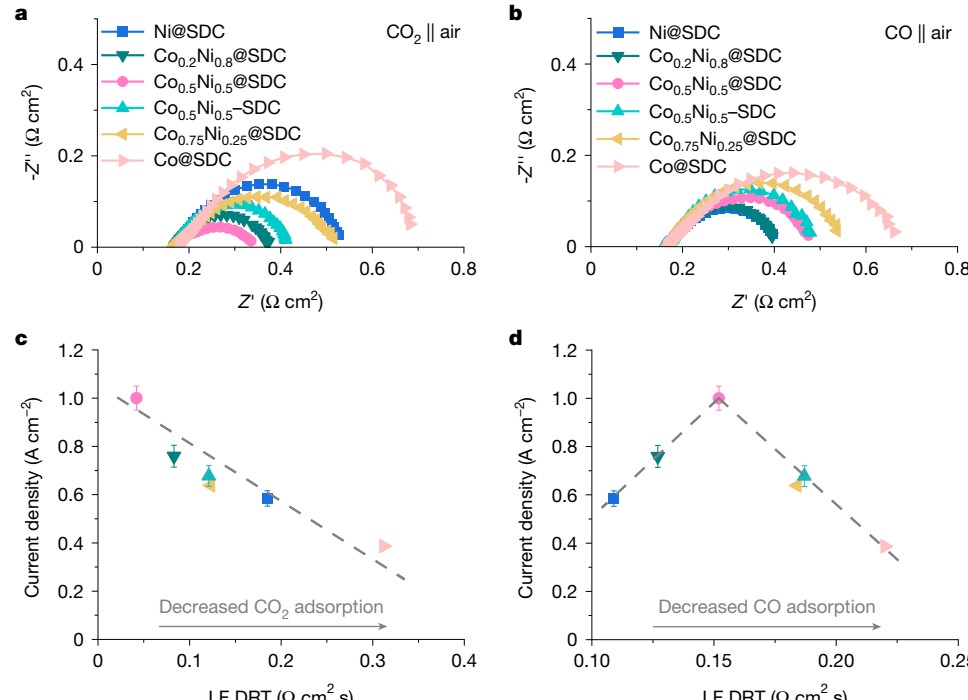

**Fig. 4 | Functioning mechanisms. a**, Nyquist plots obtained from EIS at 1.0 V under $CO_2$ atmosphere in cathode. **b**, Nyquist plots obtained from EIS at 0.9 V under CO atmosphere in anode. **c**, Relation between $CO_2$ electroreduction activity (current density at 1.1 V) and $CO_2$ adsorption and activation abilities (low-frequency intensity from $CO_2$ DRT analysis) for different catalysts. **d**, Relation between $CO_2$ electroreduction activity (current density at 1.1 V) and CO adsorption and activation abilities (low-frequency intensity from CO DRT analysis) for different catalysts.

led to low-frequency peaks under $CO_2$ first decreasing and then increasing in intensity, whereas under CO they continuously increased, with intermediate-frequency and high-frequency peaks remaining consistent (Supplementary Fig. 25). Compared with $Co_{0.5}Ni_{0.5}$–SDC, $Co_{0.5}Ni_{0.5}$@SDC exhibited smaller low-frequency peaks under both $CO_2$ and CO (Supplementary Fig. 25). Collectively, these results indicate that an optimal Co–Ni alloy enhances $CO_2$ adsorption and activation while moderating CO activation compared with monometallic Ni and Co. Compared with $Co_{0.5}Ni_{0.5}$–SDC, the encapsulated $Co_{0.5}Ni_{0.5}$@SDC had more metal–oxide interfaces; this promoted $CO_2$ adsorption and oxygen ion transport and led to higher activity. Pure SDC exhibited weak activation of both $CO_2$ and CO (Supplementary Fig. 26), consistent with its limited $CO_2$ electroreduction activity (Fig. 2a). A correlation between $CO_2$ electroreduction activity and $CO_2$ and CO adsorption and activation abilities for different catalysts showed that strong $CO_2$ adsorption and moderate CO adsorption were advantageous for $CO_2$ electroreduction in our system (Fig. 4c,d). In long-term stability tests, $Co_{0.5}Ni_{0.5}$@SDC showed modest changes in both EIS and DRT spectra, whereas other catalysts showed greater increases in $R_p$ and low-frequency peaks (Supplementary Figs. 27–29), reflecting the durability of $Co_{0.5}Ni_{0.5}$@SDC. During $CO_2$ electroreduction, coke formation proceeds by means of the Boudouard reaction: $2CO(g) \rightarrow C + CO_2(g)$[26]. Weak CO binding can also help to suppress coke formation, leading to the improved stability of $Co_{0.5}Ni_{0.5}$@SDC.

Density functional theory (DFT) simulations were conducted to determine the adsorption energies of $CO_2$ and CO on different models, as well as energy profiles for $CO_2$ electroreduction to CO. We used face-centred cubic structures with (111) and (001) terminations for the metals and a fluorite structure with a (111) termination for SDC (Supplementary Figs. 30 and 31 and Supplementary Table 10), reflecting the predominant facets observed in our catalysts (Supplementary Figs. 6 and 11). A uniform distribution of Ni and Co was assumed for the Co–Ni alloy, as no strong preference for a particular surface termination was observed, even in the presence of adsorbed CO (Supplementary Figs. 32–34). Bader

charge analysis indicated charge transfer from Co to Ni in all Co–Ni alloy models (Supplementary Table 11). $CO_2$ adsorption was weak on isolated metal or SDC surfaces, whether oxygen vacancies were present in the SDC or not, with Gibbs free energies ranging from 1.47 to 0.70 eV (Supplementary Fig. 35). This result suggests that neither the metal nor SDC alone could serve as an active site for $CO_2$ adsorption. Instead, the metal–SDC interface, represented here by $Ce_3SmO_7$ clusters adsorbed on metal surfaces, enhanced adsorption of $CO_2$ as carbonates, with Gibbs free energies ranging from −0.11 to −0.39 eV (Supplementary Figs. 35 and 36) and showing no strong correlation with the degree of charge transfer (Supplementary Fig. 37). This indicates that the metal–SDC interface is the primary active site for $CO_2$ adsorption. On the other hand, CO adsorption was more favourable on the metal sites (Supplementary Fig. 38). The choices of functional, thermal expansion effect[32] and CO coverage did not affect the overall adsorption trend (Supplementary Fig. 39 and Supplementary Table 12). Notably, Co–Ni–SDC demonstrated enhanced $CO_2$ adsorption and weakened CO adsorption compared with Ni–SDC (Supplementary Figs. 35c and 38c). On the basis of these findings, we propose a dual-site reaction mechanism for our catalysts: $CO_2$ is captured as a carbonate at the metal–SDC interface, followed by reduction of $CO_2$ to CO at the adjacent metal sites. The Gibbs free energy profiles for $CO_2$ electroreduction to CO further confirmed that the reaction was energetically favoured on the dual sites rather than on isolated metals or SDC, with similar reaction barriers on Co–Ni–SDC and Ni–SDC and little influence of oxygen coverage (Supplementary Figs. 40–46).

## Comparison with current SOECs and MEAs

We conducted a comprehensive comparison of our SOEC with state-of-the-art high-temperature SOECs and low-temperature MEAs for $CO_2$ electroreduction to CO. Our SOEC exhibited superior performance across key parameters, including lifetime, energy efficiency, CO single-pass yield and cell voltage at industrially relevant current densities (Fig. 5 and Supplementary Tables 13 and 14). The typical

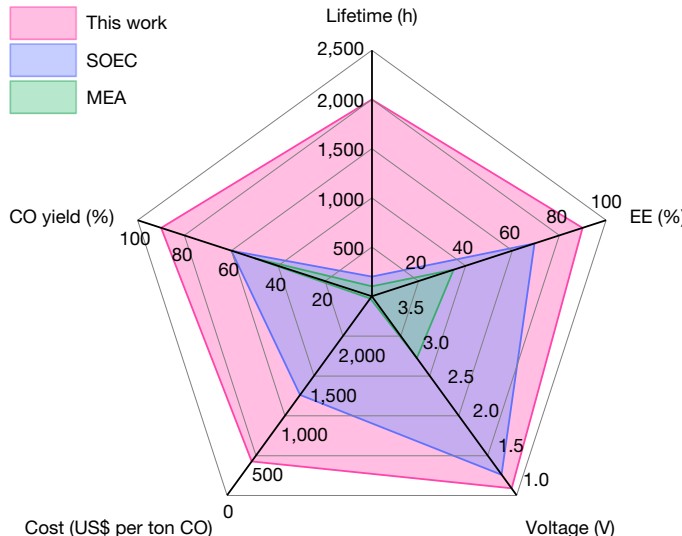

**Fig. 5 | Comparison.** Comparison of our system and state-of-the-art low-temperature MEAs and high-temperature SOECs for $CO_2$ electroreduction to CO.

lifetime, energy efficiency, CO single-pass yield and cell voltage of SOECs are less than 200 h, 70%, 60% and greater than 1.3 V respectively, at current densities of 1 A cm$^{-2}$ or higher, even for electrode-supported cells[13–21] (Fig. 5 and Supplementary Table 13). For low-temperature MEA systems, the lifetime, energy efficiency, CO single-pass yield and cell voltage were generally lower than 100 h, 35%, 50% and greater than 3.0 V, respectively, at 1 A cm$^{-2}$ or higher (Fig. 5 and Supplementary Table 14). Our SOEC achieved a lifetime, energy efficiency, CO single-pass yield and cell voltage of 2,000 h, 90%, 90% and 1.1 V, respectively, at 1 A cm$^{-2}$ for high-temperature $CO_2$-to-CO conversion (Fig. 5). As a result, on the basis of a preliminary cost estimation[16] (Supplementary Note 1 and Supplementary Tables 15–17), our SOEC demonstrated net cost reductions of approximately 60% and 80% compared with state-of-the-art SOECs and MEAs, respectively (Fig. 5). A lower electricity price (below US$0.038 per kWh), a longer lifetime, and a higher current density (up to approximately 2 A cm$^{-2}$) further contribute to the economic benefit of our system (Supplementary Fig. 47). These findings demonstrate the potential applications of our system in efficient and economically viable $CO_2$ conversion processes.

## Conclusions

In conclusion, we have successfully developed a cobalt–nickel alloy catalyst encapsulated with relatively inert SDC for high-temperature $CO_2$ electroreduction. The unique encapsulated structure, coupled with an optimized alloy composition, synergistically improves $CO_2$ adsorption, tempers CO adsorption and limits metal agglomeration, leading to both high activity and stability. This work provides a promising strategy for designing highly efficient and stable high-temperature $CO_2$ electroreduction catalysts, with potential for industrial applications.

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

## Methods

### Chemicals and materials

Nickel(II) nitrate hexahydrate (Ni(NO$_3$)$_2$·6H$_2$O, 99.9%, ABCR), manganese(II) nitrate hexahydrate (Mn(NO$_3$)$_2$·6H$_2$O, 98%, ABCR), iron(III) nitrate nonahydrate (Fe(NO$_3$)$_3$·9H$_2$O, 98%, ABCR), cobalt(II) nitrate hexahydrate (Co(NO$_3$)$_2$·6H$_2$O, 98+%, ABCR), copper(II) nitrate trihydrate (Cu(NO$_3$)$_2$·3H$_2$O, 98+%, Sigma), ammonium heptamolybdate tetrahydrate ((NH$_4$)$_6$Mo$_7$O$_{24}$·4H$_2$O, 99%, Roth), indium(III) nitrate hydrate (In(NO$_3$)$_3$·H$_2$O, 99.9%, Sigma), tin(II) chloride (SnCl$_2$, 98%, Sigma), citric acid (99.5%, Acros), ethylene glycol (99+%, Brunschwig), cerium(III) nitrate hexahydrate (Ce(NO$_3$)$_3$·6H$_2$O, 99.5%, ABCR), samarium(III) nitrate hexahydrate (Sm(NO$_3$)$_3$·6H$_2$O, 99.9%, ABCR), ethylenediaminetetraacetic acid (99%, IVALUA), ammonia solution (25%, VWR), lanthanum(III) nitrate hexahydrate (La(NO$_3$)$_3$·6H$_2$O, 99.9%, ABCR), strontium nitrate (Sr(NO$_3$)$_2$, 99+%, Fisher), LSGM (99%, Sigma), copper(II) oxide (CuO, 99%, Sigma), dispersant (2%, Fiaxell SOFC Technologies), binder (30%, Fiaxell SOFC Technologies) and carbon dioxide (CO$_2$, 99.9%, Carbagas) were all used as received without further purification.

### Synthesis of M$_x$Ni$_{1-x}$@SDC

M$_x$Ni$_{1-x}$@SDC was synthesized using a sol–gel method. First, 20 mmol of Ni(NO$_3$)$_2$·6H$_2$O and 20 mmol of a second metal salt, which included Mn(NO$_3$)$_2$·6H$_2$O, Fe(NO$_3$)$_3$·9H$_2$O, Co(NO$_3$)$_2$·6H$_2$O, Cu(NO$_3$)$_2$·3H$_2$O, Zn(NO$_3$)$_2$·6H$_2$O, (NH$_4$)$_6$Mo$_7$O$_{24}$·4H$_2$O, In(NO$_3$)$_3$·H$_2$O or SnCl$_2$, were dissolved in 50 ml of H$_2$O at 100 °C under vigorous stirring. Then, 100 mmol of citric acid and 100 mmol ethylene glycol were added to the solution to form a sol. Subsequently, 8 mmol of Ce(NO$_3$)$_3$·6H$_2$O and 2 mmol of Sm(NO$_3$)$_3$·6H$_2$O were added. The reaction was kept at 100 °C for about 3 h to evaporate H$_2$O from the system. The remaining solvent was removed at 300 °C overnight in an oven, followed by heat treatment in air at 600 °C for 5 h with a ramping rate of 5 °C min$^{-1}$. By changing the feeding molar ratio of Ni(NO$_3$)$_2$·6H$_2$O and Co(NO$_3$)$_2$·6H$_2$O while keeping their total amounts to 40 mmol, we obtained Co$_x$Ni$_{1-x}$@SDC catalysts with different molar ratios of Co and Ni. A pure SDC sample without addition of Ni(NO$_3$)$_2$·6H$_2$O or a second metal salt was also prepared.

### Synthesis of Co$_{0.5}$Ni$_{0.5}$–SDC

Co$_{0.5}$Ni$_{0.5}$–SDC without an encapsulated structure was synthesized using a step-wise sol–gel method. First, pure SDC powder was first synthesized as described. Second, 10 mmol of SDC powder was dispersed in 50 ml of H$_2$O at 100 °C under vigorous stirring, and 20 mmol of Ni(NO$_3$)$_2$·6H$_2$O and 20 mmol of Co(NO$_3$)$_2$·6H$_2$O were then added to the solution. Subsequently, 80 mmol of citric acid and 80 mmol ethylene glycol were added to form a sol. The reaction was kept at 100 °C for about 3 h to evaporate H$_2$O from the system. The remaining solvent was removed at 300 °C overnight in an oven, followed by heat treatment in air at 600 °C for 5 h with a ramping rate of 5 °C min$^{-1}$.

### Synthesis of LSCF

LSCF was synthesized using a modified sol–gel method[33]. First, 40 mmol of ethylenediaminetetraacetic acid was added to 40 ml of a 25 wt.% aqueous ammonia solution. Subsequently, 12 mmol of La(NO$_3$)$_3$·6H$_2$O, 8 mmol of Sr(NO$_3$)$_2$, 4 mmol of Co(NO$_3$)$_2$·6H$_2$O and 16 mmol of Fe(NO$_3$)$_3$·9H$_2$O were added. Next, 60 mmol of citric acid was added, and the solution pH was adjusted to 8 by addition of more aqueous ammonia solution. The resulting solution was heated at 100 °C under vigorous stirring for about 3 h to evaporate H$_2$O from the system. The remaining solvent was removed at 120 °C overnight, followed by heat treatment in air at 950 °C for 5 h with a ramping rate of 10 °C min$^{-1}$.

### Cell preparation

An electrolyte-supported cell (M$_x$Ni$_{1-x}$@SDC || SDC || LSGM || SDC || LSCF) was fabricated by a tape-casting process followed by several screen-printing processes. To prepare tape-casting slurries of the electrolyte, we mixed 20 g of LSGM powders, 14 g of dispersants and 20 g of binders by ball milling for 1 h. The resultant slurries were tape-casted into well-defined tapes with an original thickness of 1.5 mm using a tape-casting machine (MTI Corporation), followed by drying at room temperature for 48 h. The tapes were then punched into shape and sintered at 1,400 °C for 10 h with a ramping rate of 2 °C min$^{-1}$ to form a dense electrolyte support. The screen-printing slurries were prepared by mixing 2 g of powders, 2 g of dispersants and 0.8 g of binders by ball milling for 1 h. A buffer SDC layer (2 wt.% CuO was added to promote sintering[34]) was first screen-printed on to both sides of the electrolyte support using a screen-printing machine (Fiaxell SOFC Technologies) to avoid any side reaction between electrolyte and electrode. Then, the M$_x$Ni$_{1-x}$@SDC cathode and LSCF anode (35 wt.% SDC was added to the LSCF to improve O$^{2-}$ conductivity) were screen-printed on to either side of the SDC. The active areas of both cathode and anode were 1.5 cm$^2$. The printed cells were sintered at 1,200 °C for 6 h with a ramping rate of 2 °C min$^{-1}$. The thickness and mass loading of each layer are shown in Supplementary Table 1.

### Electrochemical measurements

All electrochemical measurements were conducted using a SOEC set-up from Fiaxell SOFC Technologies and were controlled by an Autolab potentiostat (PGSTAT302N) equipped with a 20-A booster. The cell was heated to 800 °C with a ramping rate of 5 °C min$^{-1}$ under air. Next, 20 vol.% H$_2$ diluted with N$_2$ was fed to the cathode compartment to reduce the catalyst for 10 min. When the reduction process was complete, CO$_2$ and air were fed to the cathode and anode compartments, respectively, at a flow rate of 50 ml min$^{-1}$. The gas outlet from the cathode compartment was connected to an online gas chromatograph (SRI Instruments) for product analysis.

The Faradaic efficiency was calculated by equation (1):

$$FE = \frac{n \times F \times x \times f_{out}}{i}, \tag{1}$$

where FE is the Faradaic efficiency; $n$ is the number of electrons exchanged to produce CO from CO$_2$ (that is, 2); and $F$, $x$, $f_{out}$ and $i$ are the Faraday constant (96,485 C mol$^{-1}$), the molar concentration of CO measured by gas chromatography (mol/mol), the flow rate of the cathode gas outlet (mol s$^{-1}$) and the current (A), respectively.

The energy efficiency was estimated by equation (2):[5]

$$EE = \frac{\Delta H \times FE}{(\text{electricity energy} + \text{heat energy})} = \frac{E_{\Delta H} \times FE}{(E_{app} + E_{\Delta S}/\eta_h)}, \tag{2}$$

where EE is the energy efficiency; and $\Delta H$, $E_{\Delta H}$, $E_{app}$, $E_{\Delta S}$ and $\eta_h$ are the enthalpy of formation, thermoneutral potential, applied cell voltage, potential based on heat energy and heating efficiency, respectively. For the reaction CO$_2$ → CO + 0.5 O$_2$ at 800 °C, $E_{\Delta H}$ and $E_{\Delta S}$ are 1.467, and 0.481 V, respectively. The $\eta_h$ used for industrial applications is usually at least 90% (refs. 35,36); thus, a value of 90% was used in our estimation.

The CO single-pass yield was calculated by equation (3):

$$\text{CO single pass yield} = x \times f_{out}/f_{CO_2}, \tag{3}$$

where $f_{CO_2}$ is the flow rate of inlet CO$_2$ gas (mol s$^{-1}$).

Operando EIS measurements were performed at 1.0 V for CO$_2$ electroreduction and 0.9 V for CO electrooxidation at 800 °C. A perturbation amplitude of 10 mV was applied with frequencies ranging from 100 kHz to 0.01 Hz. The fitting of the EIS data and DRT analysis were conducted using RelaxIS 3 software (rhd instruments).

ECSA was measured using a double-layer capacitance ($C_{dl}$) method[37], determined by means of EIS measurements using a symmetrical cell configuration (M$_x$Ni$_{1-x}$@SDC || SDC || LSGM || SDC || M$_x$Ni$_{1-x}$@SDC) at

800 °C under $N_2$ atmospheres on both sides. To establish a baseline, a flat, pure electrolyte cell without catalysts (SDC || LSGM || SDC) was fabricated. EIS data were recorded at 0 V with a perturbation amplitude of 10 mV and frequencies ranging from 100 kHz to 0.1 Hz. The electrode capacitance ($C_{dl}$) was obtained as half the capacitance derived from fitting the EIS using a simplified Randles circuit (Supplementary Fig. 5). Consequently, the ECSA of an electrode could be calculated by equation (4):

$$ECSA = C_{dl} \times S/C_{base} , \qquad (4)$$

where $C_{dl}$ represents the capacitance of a specific electrode, $C_{base}$ represents the capacitance of the baseline and $S$ denotes the geometric area of the electrode.

## Characterizations

Metal elemental analysis was conducted on a NexION 350D inductively coupled plasma mass spectrometer. X-ray diffraction patterns were recorded on a PANalytical Aeris diffractometer using Cu $K_\alpha$ radiation (40 kV, 15 mA). Scanning electron microscopy images were recorded on a Zeiss GeminiSEM 300, and the corresponding energy-dispersive X-ray spectroscopy mapping was performed at 16 kV. TEM measurements were carried out on an FEI Talos F200S at 200 kV. Spherical-aberration-corrected TEM measurements were performed on an FEI Titan Themis 60-300 instrument at 200 kV. Carbon elemental analysis was performed in an UNICUBE (Elementar) microelemental analyser by a combustion method.

XPS measurements were performed in an Omicron XPS system equipped with a pretreatment chamber, using aluminium $K_\alpha$ X-rays as the excitation source at 15 kV and 300 W. For quasi in situ XPS measurements of fresh samples, a reduction process was applied in 20 vol.% $H_2$ at 800 °C for 10 min in the pretreatment chamber. Subsequently, samples were transferred to the vacuum chamber for XPS measurements without exposure to air. After stability tests, samples underwent no further treatment but were immediately sealed in an $N_2$-protected bag.

X-ray absorption spectroscopy (XAS) measurements were performed at the 12B2 Taiwan beamline (SPring-8, Japan) of the National Synchrotron Radiation Research Center, operated at an 8.0 GeV storage ring with a constant current of approximately 99.5 mA. To prevent oxidation, all samples were sealed in an $N_2$-protected bag before XAS measurements. Measurements at the Ni K-edge (8,333 eV) and Co K-edge (7,709 eV) were performed in total fluorescence yield mode using a Lytle detector. The scan ranges were 8,133–8,933 eV for the Ni K-edge and 7,509–8,309 eV for the Co K-edge. XAS data were processed using Athena software, with energy calibration performed on the basis of the first inflection points in the absorption K-edges of the Ni and Co foils, which were set to 8333.0 eV and 7709.0 eV, respectively. Standard data processing procedures were applied, including background subtraction and edge height normalization. The average oxidation states of Ni and Co were determined by simulation of the linear combinations of reference spectra from Ni and Co foils and their oxides, NiO and CoO, respectively. Extended X-ray absorption fine structure analysis was performed using Fourier transforms on $k^3$-weighted oscillations in a $k$ range from 3.0 to 11.0 $Å^{-1}$.

Operando Raman spectroscopy measurements were performed using an inVia confocal Raman microscope (Renishaw) equipped with a 532 nm laser, coupled with a Linkam heat stage capable of controlling temperature, gas atmosphere and cell voltage. The sample was first reduced in 20 vol.% $H_2/N_2$ at 800 °C for 10 min. Then, the atmosphere was switched to $CO_2$, and Raman spectra were recorded at cell voltages of 1.2, 1.4, 1.6 and 1.8 V at 800 °C.

## DFT simulations

DFT simulations were conducted using the Vienna ab initio simulation package[38,39]. The Perdew–Becke–Ernzerhof[40] functional, supplemented with the D3 dispersion correction[41], was used to account for the van der Waals interactions. A Hubbard correction[42] was applied using the Dudarev method[43] to describe the $f$-electrons of Ce and Sm in SDC models, with a $U_{eff}$ of 4.5 and 4.0 eV for Ce and Sm, respectively[31,44,45]. Core electrons were represented using projector augmented wave core potentials[46,47], whereas valence electrons were described with plane waves at a kinetic cut-off energy between 450 and 700 eV. The Monkhorst–Pack method[48] was used to generate a $\Gamma$-centred mesh with a reciprocal grid finer than 0.036 $Å^{-1}$ for Brillouin zone sampling.

Spin polarization was included in all simulations involving Co, Ni and oxygen-defective SDC systems[45]. A kinetic cut-off energy of 450 eV was used for most simulations, except for lattice parameter optimizations, in which a higher cut-off of 700 eV was applied to avoid Pulay stress. Dipole corrections[49] were included along the $z$ axis, and a vacuum region of 15 Å was added between slabs. For Ni and Co–Ni alloys, we modelled the (111) and (001) facets as $p(4 \times 4)$ and $p(2 \times 2)$ slabs, respectively. Three different terminations were considered for the Co–Ni(001) surface. Metal slabs contained four atomic layers, with the two outermost layers allowed to relax, whereas the two bottommost layers were fixed to their bulk positions. For SDC, a $p(3 \times 3)$ slab of the $CeO_2$(111) was modelled, incorporating Sm and oxygen vacancies. This slab consisted of nine atomic layers (three O–Ce–O trilayers), with the five outermost layers allowed to relax. The Ni–SDC and Co–Ni–SDC interfaces were represented by $Ce_3SmO_7$ clusters adsorbed on $p(4 \times 4)$ slabs of Ni(001) and Co–Ni(001) surfaces, respectively. Six different $Ce_3SmO_7$ aggregates were generated on the basis of $CeO_2$(111) models, in which a Ce atom was replaced by Sm, and an oxygen vacancy was introduced. Various adsorption configurations of these clusters were then explored on the metallic surfaces, resulting in 18 and 24 different structures on Ni(001) and Co–Ni(001), respectively.

For the adsorption energy of $CO_2$ and CO and the energy profiles for $CO_2$ electroreduction, $CO_2$ and CO molecules in the gas phase were used as thermodynamics references. Vibrational contributions to enthalpy and entropy were included in the Gibbs free energies calculations, along with rotational and translational contributions for gas-phase $CO_2$ and CO. Gibbs free energies were computed at 800 °C and 1 atm of $CO_2$, in accordance with the experimental reaction conditions. Transition states were located using the climbing image nudged elastic band method[50]. Numerical frequencies calculations, using a step size of ±0.015 Å, were performed to confirm the nature of the transition states. Vibrational contributions to enthalpy and entropy for the Gibbs free energies of transition states were computed using standard approximations: the ideal gas model, rigid rotor and harmonic oscillator. Only the reactants were considered for metals, whereas for oxidic systems, oxygen atoms were also included in the vibrational estimations owing to their lower molecular weight.

## Data availability

All data are available in the article and its Supplementary Information. Data for all experiments and DFT calculations are available at Zenodo (https://doi.org/10.5281/zenodo.14540651)[51] and ioChem-BD (https://doi.org/10.19061/iochem-bd-1-314)[52–54], respectively. The atomic positions of all DFT models are provided in the Supplementary Data. Source data are provided with this paper.

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

**Acknowledgements** W.M., J. Tian, W.R. and X.H acknowledge the financial support of EPFL. J.M.-V. and N.L. acknowledge financial support from the Spanish Ministry of Science and Innovation (PID2021-122516OB-I00 and Severo Ochoa Grant MCIN/AEI/10.13039/501100011033 CEX2019-000925-S), Generalitat de Catalunya and AGAUR (2023 CLIMA 00105). We thank Barcelona Supercomputing Center-MareNostrum (BSC-RES) for providing generous computational resources. H.M.C. acknowledges financial support from the National Science and Technology Council, Taiwan (contract no. NSTC NSTC 112-2123-M-002-008) and from National Taiwan University (NTU-CC-113L893304). S.J. and J.L. acknowledge financial support from the European Research Council (ERC) under the European Union's Horizon 2020 research and innovation programme (grant: CATACOAT, no 588251). J. Taubmann and C.C acknowledge financial support from Innovation Fund Denmark (1150-00001B); further information is available at www.MissionGreenFuels.dk. We thank J. Cai and G. Yang from ShanghaiTech University for conducting some XPS measurements using the BL02B01 of the Shanghai Synchrotron Radiation Facility and SPECS ambient pressure XPS instrument with financial support from the National Natural Science Foundation of China (no. 11227902); G. Wang, R. Li and Y. Shen from the Dalian Institute of Chemical Physics for assistance with some characterizations; and B. Victor from CIME, EPFL, for conducting aberration-corrected high-angle annular dark-field scanning transmission electron microscopy.

**Author contributions** W.M. designed and synthesized the catalysts, and performed most of the electrochemical tests, characterizations and analyses; J.M.-V. performed the DFT simulations and drafted the DFT parts of the paper under the supervision of N.L.; J. Tian performed operando Raman measurements under the supervision of J. Taubmann and C.C; M.-T.L. performed the XAS measurements under the supervision of H.M.C.; S.J., W.R. and J.L. assisted with some measurements. W.M. and X.H. wrote the paper, with input from all the other authors. X.H. directed the research.

**Funding** Open access funding provided by EPFL Lausanne.

**Competing interests** W.M. and X.H. are inventors on a European patent application (PCT/EP2024/069715) that includes the catalyst described in this work. The other authors declare no competing interests.

**Additional information**
**Correspondence and requests for materials** should be addressed to Xile Hu.
