## [Peer Review File · Nature]

Encapsulated Co-Ni alloy boosts high-temperature CO₂ electroreduction

Corresponding Author: Professor Xile Hu

Version 0:

Reviewer comments:

Referee #1

(Remarks to the Author)

In this manuscript, the authors report a Co-Ni alloy catalyst encapsulated by Sm₂O₃-doped-CeO₂ for high-temperature CO₂-to-CO conversion. The catalyst exhibits a high energy efficiency of 90%, a lifetime of up to 2000 hours, a single-pass yield of 90%, and a selectivity for CO of around 100% at a current density of 1 A/cm². It is the fact that the catalyst largely extends the lifetime of high-temperature CO₂ electroreduction in solid oxide electrolysis cell from 1000 hours to an impressive 2000 hours, whereas the mechanism of the long-term stability has not been well investigated and explained, which may not provide directional guidance for the design of highly stable catalysts. Furthermore, the electroreduction of CO₂ in a solid oxide electrolysis cell is not novel and there is no groundbreaking research presented in the manuscript. The reviewer recommends enhancing the suitability for further publication by proposing the following suggestions:

1. The reported catalyst exhibits good catalytic activity and stability. To give useful information for further exploring advanced catalysts, I recommend the authors to use DFT calculations and in situ physicochemical characterizations such as X-ray photoelectron spectroscopy, X-ray adsorption spectroscopy and electrochemical characterization such as AC impedance spectroscopy and distribution of relaxation times analysis to monitor the elementary processes of the electrode reactions, as well as the degradation processes of the SOEC during long-term operation.
2. The CoNi@SDC catalyst should be carefully studied after a 2000-hour stability test. This will enable a comprehensive understanding of whether the catalyst has degraded, for examples, if there is any mass loss? Whether the atomic ratio of Co: Ni has been altered? If not, what is the cause of the observed stability? After all, Co-Ni alloy is a very common catalyst that has been intensively studied before.
3. Regarding the structure of Co-Ni alloy encapsulated in SDC, the authors have investigated it by SEM-EDX and TEM-EDX in the main text and supplementary information (Fig. 3d, Supplementary Figs. 6a and 7a). However, the reviewer think that relying solely on these data can only show that Co-Ni alloy and SDC indeed mixed together. These data cannot offer conclusive evidences that Co-Ni alloy encapsulated in SDC. Similarly, the Ni encapsulated in SDC cannot be confirmed solely by the characterization presented in the manuscript. Further characterizations, such as spherical aberration corrected transmission electron microscopy, are required.
4. On page 8 lines 6-8 of the main text, the authors conclude that CoNi@SDC has the effect of suppressing metal agglomeration by comparing the SEM-EDX and TEM-EDX of CoNi@SDC, CoNi-SDC and Ni@SDC before and after the test, but this proof is still not convincing enough. It is recommended to supplement the particle size distribution evolution of the three samples before and after the reaction to prove the effect of CoNi@SDC.
5. Supplementary Fig. 11, the authors prove that CoNi@SDC has the effect of suppressing coke formation. However, Raman testing is the statistical analysis of surface signals in a certain area at the micro-scale. The results are potentially accidental because it is possible to select areas that do not have carbon deposition. Carbon sulfur analyzer maybe offer a more objective and accurate statistical method for carbon content detection. I suggest the author to supplement this data.
6. In the CoNi@SDC catalyst, is the metal active site Co or Ni, or both? If the active site is a single metal, what role does the other metal play in the CoNi@SDC catalyst? The author should present rigorous experimental results to clarify this aspect.
7. The CoNi@SDC catalysts showed no discernible morphological alterations after 2,000 h of electrolysis, whereas the Ni@SDC catalysts experienced significant structural changes after 500 h of electrolysis, exhibiting obvious metal agglomeration. Given that Ni metal encapsulated within the SDC underwent significant agglomeration, this strongly suggest that the encapsulation structure cannot prevent metal agglomeration. It is therefore that the Co-Ni alloy composition plays a significant role in inhibiting agglomeration and coking formation in the CoNi@SDC catalyst. How the alloy component with a Co: Ni ratio of 0.5:0.5 inhibits metal agglomeration and coking formation? How about other Co: Ni ratios?

8. TPD measurements showed that the CoNi@SDC catalysts exhibited higher temperatures for CO₂ desorption compared to Ni@SDC at temperature ≥ 500 °C, while lower temperatures were observed for CO desorption in this temperature range. This suggests that the Co-Ni alloy enhanced the adsorption of CO₂ but weakening the adsorption of CO when compared to pure Ni in this temperature range. Thus, the authors attributed the higher activity of Co-Ni alloy to stronger CO₂ adsorption and weaker CO binding. However, the evaluation of CO₂ electroreduction performance is conducted at a temperature of 800°C, which is significantly higher than the temperature range of the TPD measurements. The reviewer reasonably believe that the TPD results cannot accurately represent the strength of CO/CO₂ adsorption by the catalyst at the actual reaction temperature of 800°C.

Referee #2

(Remarks to the Author)

The authors have developed an encapsulated Co-Ni alloy electrocatalyst which exhibited good performance towards CO₂RR to produce CO, especially excellent in activity and stability at high current of 1 A/cm². They explained the good electrocatalytic performance for CO₂ reduction were attributed to the enhanced CO₂ adsorption, weaken CO adsorption and suppressed metal agglomeration. The manuscript demonstrates a commendable level of originality by introducing a facile approach to synthesize high-efficient and durable CO₂ reduction catalyst. Moreover, the findings presented in this work hold promise for large-scale CO₂ conversion to CO, underscoring the manuscript's high level of significance. Overall, this manuscript demonstrates a solid foundation of research and addresses an important topic in the field. The methodology is well-structured and the data analysis is thorough. However, there are several areas that need attention before considering it for publication.

The normalized activity based on BET surface areas showed interesting results, however, for the electrocatalytic activity, the normalized activity based on electrochemically active surface area (ECSA) is more convincing than specific surface area. What are the ECSA normalized activities? Also, the BET surface area normalized activities of CoNi-SDC \approx CoNi@SDC, which means that these two catalysts have the same intrinsic activity? The author should further explain the role of encapsulation. Does the metal oxides actually block the active sites (CoNi) by encapsulation? Then the author actually did trade the activity with the stability.

In terms of the stability, I don't think the CoNi-SDC experienced a significant increase in cell voltage from 1.21 to 1.51 V after 500 h of operation, as the author stated. Within the first several hours, the voltage increased rapidly from 1.21 to 1.51 V, which is obviously a potential increase that is a normal phenomenon in the chronopotentiometry curve. The potential of CoNi-SDC stabilized at 1.51 within 500 h. To me, CoNi-SDC is stable and it is important to know what happens after 500 h to 2000 h. If the activity drops, we can conclude that encapsulation can inhibit the activity drop, otherwise such a conclusion is not supported by reliable data evidence.

The author concluded that the Co-Ni alloy enhanced the adsorption of CO₂ while weakening the adsorption of CO when compared to pure Ni in this temperature range, then what makes CoNi@SDC better than CoNi-SDC when is has similar CO₂ and CO adsorption/desorption ability? Is this reasonable? Because CO tends to adsorb on metal sites (Ni-Co) which are encapsulated by metal oxides. The observed decrease in CO adsorption in CoNi@SDC is more reasonable. So far, it is clear to me why CoNi@SDC and CoNi-SDC show better activity and stability than Ni@SDC, but it is not clear why CoNi@SDC > CoNi-SDC because the results show that CoNi-SDC with good stability (data until 500h) and intrinsic activity similar to encapsulated CoNi@SDC.

Are there interactions (electronic and structural) between CoNi and SDC and between Co and Ni? It is not clear how the presence of these two phases in the system. XPS and/or XAS are necessary to study the electronic structures of the catalysts.

For the statement in the introduction: "Because the reaction is conducted in a gas-phase process with pure CO₂ as reactant (without water), a complete selectivity for CO formation can be obtained." It is better to explain why 100% CO selectivity in gas phase and cite reference if necessary.

Figure 1b, you should reconsider naming the left block instead of "Current catalysts". This is very general term. Do you mean that the current existing catalysts are all metal, metal+oxide, and oxide-supported metal? And encapsulated alloy in this work developed as a new term for the first time? Apparently that is not the case, see references: ACS Catal. 2023, 13, 8, 5135–5146 and Applied Catalysis B: Environmental, 283, 2021, 119628.

Since the title emphasized high-temperature CO₂ reduction, should the author provide the reaction temperatures for their performance data, for both ref. [4-8] and data from this study.

Referee #3

(Remarks to the Author)

Review of "Encapsulated Co-Ni alloy boosts high-temperature CO₂ electroreduction"

This is an interesting paper that proposes a new type of catalyst that boosts high temperature electroreduction. Overall, the paper is quite interesting and certainly has a significant amount of intellectual merit. The results also seem to be significant since the improved catalytic activity and the stability of the catalyst under consideration is quite significant. The data also seems to be of high quality for the presented work.

However, the referee does have two concerns associated with the paper that should be addressed before it can be accepted to such a prestigious journal:

1. The results shown in Figure 2 are certainly interesting. However, in order to better understand the synergy between the two metals it would be also interesting to try other cases such as Co@SDC. Is there any particular reason why this cannot

be done?

2. With regard to the results that are shown in Figure 3, the referee is unsure about the results that are shown and in particular what does the low temperature peak correspond to (in the 100 to 200 degree Celsius range). Some discussion in the text would be useful here.

3. How should the reader understand the high temperature peak? This would correspond to a very unusually high binding energy of carbon dioxide. To have this paper published in such a prestigious publication as Nature, the referee thinks that there should be some sort of first principles-based calculations that would be reported in the manuscript. In this way, the reader would be better able to rationalize the results.

Version 1:

Reviewer comments:

Referee #1

(Remarks to the Author)

The reviewer appreciates the responses from the authors that have addressed most of my concerns. The manuscript is greatly improved, but there are still some suggestions that the author may want to consider:

1. According to previous research (Angew. Chem. Int. Ed. 2024, 63, e202313361), when CO₂ electrolysis is performed in SOEC, CO₂ molecules will first be adsorbed on the cathode surface to form a carbonate intermediate, and then the carbonate intermediate will accept electrons and dissociate into CO and O²⁻ at the three phase boundaries. Finally, CO is desorbed from the cathode surface, O²⁻ is transported to the anode through the electrolyte, and O₂ molecules are generated through the oxygen evolution reaction. Is it possible that the authors obtain a volcano-shaped curve about the CO₂ adsorption and activation and CO desorption of the different catalysts you tested? As such, this will provide more mechanistic guidance for audience to design catalysts.

2. It is recommended that the authors supplement the in-situ NAP-XAS test to detect the intrinsic reaction mechanism during carbon dioxide electrolysis. The difference in the production of key intermediates between different catalysts can be reflected by capturing the signals of carbonate species, thereby providing more comprehensive understandings for the catalytic performance differences.

3. Regarding the techno-economic analysis, the author mainly calculated the operating costs. Is it possible to provide additional calculations on how much economic benefits there would be if the device is used for commercial and stable operation to generate CO? This will help with evaluating the future development trend of high-temperature CO₂ electrolysis.

Referee #2

(Remarks to the Author)

I have reviewed the revised manuscript and appreciate the substantial additional experiments conducted, including DFT calculations, in-situ XPS, XAS, EIS, and DRT analyses. These efforts significantly strengthen the manuscript by providing robust evidence of the excellent performance of the Co-Ni alloy catalyst encapsulated by Sm₂O₃-doped CeO₂ for the CO₂ to CO reaction.

However, I have some inquiries regarding the experimental analysis:

XAS Analysis: It appears that fitting analysis for the XAS data has not been performed. Fitting analysis is essential for quantitatively determining the oxidation state, coordination environment, and electronic structure of the catalytic materials. This information is crucial for understanding the active sites and configuration of the materials. Could the authors provide a specific reason for this omission?

EIS Results: I noticed that the EIS results were presented, but no fitting analysis was conducted on these data. Fitting the EIS data could provide insightful information regarding the charge transfer resistance, double-layer capacitance, and other electrochemical parameters that are vital for interpreting the catalytic behavior in electrochemical systems. I suggest that the authors perform and discuss the fitting of the EIS data to better elucidate the electrochemical properties of the catalyst under study.

In Table S12, the manuscript mentions a 2000-hour lifetime for the catalyst and membrane. Could the authors clarify how this figure was determined? Is it based on experimental data or extrapolated from shorter-term tests? And in Table S13, the cost of catalyst is as low as 21 \$/ton, did the author consider the complex synthesis procedure of core-shell structured catalysts (capital cost and operational cost) or only the cost of raw materials are taken into account.

By addressing these points, the manuscript could provide a more comprehensive understanding of the catalyst's behavior, contributing further to the field.

Referee #3

(Remarks to the Author)

Reviewer 3 has re-reviewed the manuscript. The referee appreciated the additional information that was put into the manuscript to address comment 1. The referee does understand why the results for the TPD were removed.

The new DFT-based results are not compelling, however, for several reasons:

1. The authors seem to assume a uniform distribution of Ni and Co in the lattice in the models that are considered. It is well-known that adsorbates can induce segregation effects where one of the metals that the adsorbate binds to will “pump” the metal atom to the surface. This seems to have not been considered by the authors and it should have been in order to have compelling results.
2. More importantly, the adsorption energy of CO₂ is tiny (not more than -0.6 eV) and will not remain on the surface during the experiments at 800 degrees. It would seem to the referee that the authors will need to consider the effect of the SDC on the adsorption energies. The referee would anticipate that the CO₂ would form a carbonate that would anchor the CO₂ to the surface. The referee believes that a more realistic model should be considered in the underlying reaction mechanism.
3. The authors mention that “Spin polarization was included when needed.” Could the authors be more specific on this? Since Co is magnetic, the referee would anticipate that all calculations involving Co should be spin polarized. If this is indeed the case, this should be mentioned explicitly.
4. The lattice parameters should be given for all the calculations. In particular, since Co is hexagonal closed packed, there should be more than one parameter when giving the lattice constants. This information is needed if the results are to be reproducible.
5. The authors mention that the NEBs were calculated. The actual NEBs should be given in the supplementary information.

Referee #4

(Remarks to the Author)

The manuscript focuses on the synthesis of an encapsulated Co-Ni alloy catalyst doped with Sm₂O₃ and CeO₂ for high-temperature electrochemical CO₂ reduction. As noted, electrochemical CO₂ reduction is often limited by energy efficiency and the catalyst's limited lifetime at high temperatures. The superior catalytic performance of this structure is characterized and rationalized through both experimental and theoretical methods, presenting an efficient strategy to overcome the typical trade-off between activity and stability.

As a theorist, I concentrate my review on the computational aspects of the paper. Density Functional Theory (DFT) calculations are applied to determine the adsorption energies of CO₂ and CO, as well as the energy profiles for CO₂ electroreduction to CO. I have several important concerns about this section. The theoretical calculations section appears to be relatively undetailed, and more calculations need to be provided relative to the experimental work to provide a theoretical basis for the experimental findings. Specifically, the choice of computational methods and structures lacks proper justification, and the conclusions drawn from the calculations are not well-supported.

1. According to Fig. 3e, Co and Ni are randomly distributed; however, in the DFT-calculated structures, Co and Ni are arranged alternately. The authors should consider a wider range of Co and Ni combinations in their calculation model. As a typical core-shell structure, the Sm₂O₃-doped-CeO₂ (SDC) likely plays a critical role in the stability of the Co-Ni alloy and may influence catalytic performance, yet it is entirely ignored in the current version. Considering the superior catalytic performance, it is suggested to conduct the DFT calculations based on a more realistic structural model.
2. The computational methods should be described explicitly to enable reproduction by other researchers. In the current version, spin polarization is not mentioned. Additionally, there is inconsistency in the plane-wave cutoff values: 450 eV is mentioned on line 5 of the DFT simulation section (Page 17), while 700 eV is noted on line 8, which is confusing. Furthermore, the validity of the computational methods is under question. For the Co-Ni system, the commonly used PBE functional may not correctly describe the correlation interactions. More accurate methods employing different functionals should be considered in this calculation.
3. The coverage of molecules can significantly impact the calculated results. In electrochemical calculations, Gibbs free energy is commonly used rather than adsorption energy, as it considers vibrational frequency and entropy. Additionally, thermodynamic corrections are not included in the calculation of the CO₂ reduction profile (Fig. S31).
4. As shown in Fig. S29, the adsorption of CO is much stronger than that of CO₂. What then is the driving force to displace the adsorbed CO molecules? Furthermore, the adsorption energies of CO₂ on CoNi(001) are -0.48 and -0.40 eV, while on Ni(001), it is -0.43 eV. These results do not convincingly illustrate enhanced CO₂ adsorption.

Version 2:

Reviewer comments:

Referee #1

(Remarks to the Author)

The authors have resolved all my concerns. I recommend the publication of this manuscript as it is.

Referee #2

(Remarks to the Author)

I have reviewed the revised manuscript and am satisfied with the changes made by the author. I agree that the manuscript is ready for publication in its current form.

Referee #3

(Remarks to the Author)

The referee appreciates that authors have done additional calculations for the new version of the paper. However, the referee still has concerns with the revised manuscript. As such, the theoretical calculations are still not that convincing. The following points were adequately addressed by the authors:

1. Point 3 and 4 of referee 3 were adequately addressed by the authors.

The following points should be addressed:

2. The referee does appreciate that the authors have done some additional calculations regarding the segregation energies, but there needs to be more details in order for the calculations to be reproducible since the configurations that the authors have used to calculate these values are not given in the associated document. All the associated structures should be given.

3. Although the referee does appreciate the efforts that the referees have made on point 2. The referee still has some concerns regarding the revised calculations.

(i) It is somewhat unclear how the structures that are depicted in Supplementary Figure 32 c were obtained. Indeed, it seems somewhat arbitrary how the structure was chosen. Also, it is mentioned in the caption that other structures were considered but only the most stable are shown. Those structures should be shown in the text.

(ii) The referee has two concerns regarding the Gibbs free energy calculations. First, the Gibbs free energies are computed at a given temperature at pressure. The values of the temperature and the pressure are not given in the SI. This essential detail should be given.

(iii) It is also unclear where the vacancies lie in Supplementary Figure 33 b. This need to be clearly shown in the figure.

(iv) It does not make any sense that an adsorption energy to the Gibbs free energies will change by 1.5 eV. There needs to be more details in the manuscript why we should have such a large change.

4. Regarding point 5. The referee has multiple concerns with what is presented in Figure 36. In the original review, the referee requested that the NEBs be shown. What is shown in Figure 36 is not the NEBs. There should be several images between the initial state, the transition state and the final state. Also, why are there no barriers that are computed in panel b and c of this figure? In addition, the Gibbs free energies are positive in panel a and b in the figure, which means that the process would not happen. It does not really make sense to show this. As for the process shown in panel c of this figure, the CO₂ is very weakly bound to the surface so that this process is quite unlikely, too.

A general comment: The authors claim that the active site is the one where we have an adsorption energy of about 2 eV. That could also mean that the CO₂ is too strongly adsorbed on the surface so that the surface would get poisoned with CO₂. This should be mentioned in the manuscript. As such, the discussion should be moderated.

Referee #4

(Remarks to the Author)

The authors have included additional DFT calculations in the revised manuscript, detailing the calculations, structural motifs, and adsorption energy analysis. These enhancements have improved the quality of the work. However, there are a few suggestions for the DFT section:

O1. The decomposition of CO₂ on the surface generates oxygen, which can accumulate and passivate active sites. It's crucial to investigate how varying oxygen coverage affects reaction activity.

2. The NEB calculation is shown in SI Fig. 36. Activation barrier energy values should be included. Results suggest that the Ni(001) surface outperforms CoNi(001), with CoNi(001) performing comparably to itself. A detailed discussion of these NEB results is needed. Additionally, some calculation details are missing, such as thermal corrections for the NEB results and the calculated vibrational frequencies. The substrate may also influence NEB outcomes.

3. The adsorption of CO₂ and CO is affected by temperature and the reactants' partial pressure, which has been overlooked in the current calculations.

4. The authors should discuss the relationship between enhanced CO₂ adsorption and charge transfer, as this could provide valuable insights for material design.

Version 3:

Reviewer comments:

Referee #4

(Remarks to the Author)

The manuscript has been substantially enhanced through the inclusion of additional calculations and discussions. It effectively demonstrates that both alloy composition and encapsulated structure are pivotal in determining the activity and stability of catalysts in high-temperature electrochemical CO₂-to-CO reactions. Upon thorough review of the responses to previous comments, the structural models now appear reasonable and reproducible, aligning well with the experimental results. The observed variations in activity across different structural models are particularly noteworthy, suggesting that CO₂ and CO may adsorb on two distinct sites rather than a single site. Although the synergistic interactions are not fully explored in the current approach, this version of the manuscript is thought-provoking and merits publication as is, as it paves the way for further research in this area.

Response to referees

We would like to thank the three referees for the comments and suggestions. The manuscript has been subjected to a major revision according to their comments. In particular, we have conducted density functional theory (DFT) simulations and additional experimental work, including quasi in-situ X-ray photoelectron spectroscopy (XPS), X-ray absorption spectroscopy (XAS), operando Raman, operando electrochemical impedance spectroscopy (EIS), and distribution of relaxation time (DRT) analyses. Below is a description of our point-by-point responses to the comments and the changes that have been made. The changes in the main text and SI have been highlighted.

Referee #1.

General Comment.

In this manuscript, the authors report a Co-Ni alloy catalyst encapsulated by Sm₂O₃-doped-CeO₂ for high-temperature CO₂-to-CO conversion. The catalyst exhibits a high energy efficiency of 90%, a lifetime of up to 2000 hours, a single-pass yield of 90%, and a selectivity for CO of around 100% at a current density of 1 A/cm². It is the fact that the catalyst largely extends the lifetime of high-temperature CO₂ electroreduction in solid oxide electrolysis cell from 1000 hours to an impressive 2000 hours, whereas the mechanism of the long-term stability has not been well investigated and explained, which may not provide directional guidance for the design of highly stable catalysts. Furthermore, the electroreduction of CO₂ in a solid oxide electrolysis cell is not novel and there is no groundbreaking research presented in the manuscript. The reviewer recommends enhancing the suitability for further publication by proposing the following suggestions:

Our response: Thank you for the feedback and suggestions. We now provide additional experimental and theoretical data to understand the origin of the improved stability, and to also provide guidance for catalyst design. The details are described in our responses to the specific comments of the referee below.

Furthermore, we clarify two major breakthroughs of our work: 1. We demonstrate 10-time improvements in the lifetime while increasing 30% the energy efficiency, compared to the previously best records for industrially relevant high-temperature CO₂ electroreduction. That is, we have advanced the state of the art from 200 hours at 70% energy efficiency to 2,000 hours at 90% energy efficiency at 1 A/cm². 2. We show that the superior efficiency is due to a unique encapsulated structure and an optimized alloy composition. Therefore, our work not only achieves a milestone in the development of CO₂ electroreduction, but also provides a new strategy in catalyst design for high-temperature reactions, which overcomes the typical trade-off between efficiency and stability.

Specific Comments.

Comment 1: The reported catalyst exhibits good catalytic activity and stability. To give useful information for further exploring advanced catalysts, I recommend the authors to use DFT calculations and in situ physiochemical characterizations such as X-ray photoelectron spectroscopy, X-ray adsorption spectroscopy and electrochemical characterization such as AC impedance spectroscopy and distribution of relaxation times analysis to monitor the elementary processes of the electrode reactions, as well as the degradation processes of the SOEC during long-term operation.

Our response: We appreciate your insightful recommendations to improve the fundamental understanding of our catalysts. We have now added density functional theory (DFT) simulations and additional experimental data, including quasi in-situ X-ray photoelectron spectroscopy (XPS), X-ray absorption spectroscopy (XAS), operando electrochemical impedance spectroscopy (EIS), and distribution of relaxation time (DRT) analyses. These new data offer valuable insights into the elementary processes of electrode reactions and the degradation mechanisms of the SOEC during prolonged operation.

We have added the following paragraphs in the revised manuscript to describe the above results:

“Density functional theory (DFT-PBE-D3) simulations were conducted to determine the adsorption energies of CO₂ and CO, as well as the energy profiles for CO₂ electroreduction to CO, on Co-Ni alloy and Ni models, as depicted in Supplementary Figs. 28-31. Slabs with (111) and (001) terminations were utilized, reflecting the predominant facets in our catalysts (Supplementary Fig. 6). Bader charge analysis indicated a charge transfer from Co to Ni in all Co-Ni alloy models (Supplementary Table 9). In Comparison to Ni surfaces, Co-Ni alloy demonstrated either enhanced or comparable CO₂ adsorption, particularly evident on models featuring Co atoms on the outermost layer (Supplementary Fig. 29). In contrast, CO adsorption was weaker on Co-Ni alloy models across all facets compared to pure Ni (Supplementary Fig. 29). These trends persisted even when considering metal thermal expansions at 800 °C (Supplementary Fig. 30).²⁹ Furthermore, the computed energy profiles for CO₂ electroreduction showed that Co-Ni alloy exhibited a lower reaction barrier compared to pure Ni (Supplementary Fig. 31). Consistent with DRT analysis, these DFT simulations indicate that Co-Ni alloy enhances CO₂ adsorption while weakens CO adsorption relative to Ni, thereby contributing to its superior CO₂ electroreduction to CO performance.”

“The quasi in-situ XPS spectra of Ni 2p (Supplementary Fig. 13a) and Co 2p (Supplementary Fig. 13b) unveiled a reduction in the binding energy of metallic Ni in Ni@SDC and Co in Co@SDC compared to their pure forms, respectively. Meanwhile, the Ce 3d XPS spectra (Supplementary Fig. 13c) revealed a decreased Ce³⁺ content for both samples compared to pure SDC. These trends suggest a charge transfer from Ce³⁺ to either Ni or Co within each sample. This phenomenon persisted upon alloying, with the oxidation states of Co and Ni maintained below zero, and a slight increase in Ce³⁺ content observed compared to their monometallic counterparts. Such alterations are conducive to

enhanced oxygen-ion transport.⁴ Moreover, $\text{Co}_{0.5}\text{Ni}_{0.5}@SDC$ showed an enhanced charge transfer between Co-Ni and SDC compared to $\text{Co}_{0.5}\text{Ni}_{0.5}\text{-SDC}$, suggesting an increased presence of metal-oxide interfaces through encapsulation. Consistent with XPS findings, X-ray absorption spectroscopy (XAS) and operando Raman spectra corroborated the observed charge transfer from oxide to metal in all samples, with $\text{Co}_{0.5}\text{Ni}_{0.5}@SDC$ demonstrating more metal-oxide interfaces than $\text{Co}_{0.5}\text{Ni}_{0.5}\text{-SDC}$ (Figs. 3g, 3h, Supplementary Figs. 14,15 and Tables 4,5).” “Post-stability XPS and XAS spectra revealed severe Co oxidation in $\text{Co}@SDC$, contrasting with modest changes in the oxidation states of Co and Ni in Ni-containing catalysts (Supplementary Figs. 22 and 23), likely due to the higher oxygen affinity of Co relative to Ni.²⁷”

“To understand catalyst performance variations, operando electrochemical impedance spectroscopy (EIS) measurements and distribution of relaxation time (DRT) analyses were conducted under CO_2 and CO atmospheres. The EIS data delineated a trend wherein the polarization resistance (R_p) decreased in the sequence of $\text{Ni}@SDC > \text{Co}_{0.5}\text{Ni}_{0.5}\text{-SDC} > \text{Co}_{0.5}\text{Ni}_{0.5}@SDC$ under CO_2 (Fig. 4a), and $\text{Co}_{0.5}\text{Ni}_{0.5}\text{-SDC} > \text{Co}_{0.5}\text{Ni}_{0.5}@SDC > \text{Ni}@SDC$ under CO (Fig. 4b). Meanwhile, ohmic resistances (R_o) remained consistent. Regarding DRT analysis, it is established that the high-frequency (HF), intermediate-frequency (IF), and low-frequency (LF) peaks correspond to oxygen-ion migration in the electrolyte, surface oxygen transfer in the air side, and electrochemical adsorption and activation processes in the fuel side, respectively.^{17,28} The LF peaks decreased in $\text{Ni}@SDC > \text{Co}_{0.5}\text{Ni}_{0.5}\text{-SDC} > \text{Co}_{0.5}\text{Ni}_{0.5}@SDC$ under CO_2 , and $\text{Co}_{0.5}\text{Ni}_{0.5}\text{-SDC} > \text{Co}_{0.5}\text{Ni}_{0.5}@SDC > \text{Ni}@SDC$ under CO, while IF and HF peaks remained consistent (Figs. 4c and 4d). Collectively, our results indicate that Co-Ni alloy enhances CO_2 adsorption and activation but weakens CO activation compared to Ni, with more presence of metal-oxide interfaces facilitating the activation processes of both CO_2 and CO. Pure SDC exhibited weak activation of both CO_2 and CO (Supplementary Fig. 24), consistent with its limited CO_2 electroreduction activity (Fig. 2a). During the long-term stability tests, $\text{Co}_{0.5}\text{Ni}_{0.5}@SDC$ displayed modest changes in both the EIS and DRT spectra, whereas other catalysts experienced higher increases of R_p and LF peaks (Supplementary Figs. 25-27), underscoring the durability of $\text{Co}_{0.5}\text{Ni}_{0.5}@SDC$.”

In summary, the incorporation of DFT calculations, quasi in-situ XPS, XAS, EIS and DRT analyses enriches our understanding of the catalytic mechanisms and degradation processes occurring within the SOEC system. These findings contribute valuable insights for the development of highly active and stable catalysts in future research endeavors.

Comment 2: The $\text{CoNi}@SDC$ catalyst should be carefully studied after a 2000-hour stability test. This will enable a comprehensive understanding of whether the catalyst has degraded, for examples, if there is any mass loss? Whether the atomic ratio of Co: Ni has been altered? If not, what is the cause of the observed stability? After all, Co-Ni alloy is a very common catalyst that has been intensively studied before.

Our response: As you suggested, we have rigorously characterized the $\text{Co}_{0.5}\text{Ni}_{0.5}@$ SDC catalysts after the stability test. Our findings indicated that there were no discernible changes in mass loss or alterations in the atomic ratio of Co and Ni within $\text{Co}_{0.5}\text{Ni}_{0.5}@$ SDC (Supplementary Tables 6 and 8). However, we observed a modest increase in the average particle size of $\text{Co}_{0.5}\text{Ni}_{0.5}$ and SDC components in $\text{Co}_{0.5}\text{Ni}_{0.5}@$ SDC, with increases of about 4.9% and 2.5%, respectively (Supplementary Figs. 19a and 20a). Consequently, there was a slight increase in R_p and LF peak (Supplementary Figs. 25-27). Additionally, there was a slight increase of ohmic resistance (Supplementary Figs. 25-27), indicating some degradation in the electrolyte. Therefore, we attribute the modest increase in average particle size and electrolyte degradation to the slight degradation of the $\text{Co}_{0.5}\text{Ni}_{0.5}@$ SDC-based cell during a 2000-hour stability test.

It should be mentioned that conventional Co-Ni alloy catalysts, either in the form of pure metals or a simple mixture of metals and oxides, suffer from limited active interfaces and severe particle agglomeration. Through a unique encapsulation strategy combined with optimized alloy composition, now we make the Co-Ni alloy catalyst highly efficient and stable for high-temperature CO_2 electroreduction. We believe that our findings provide valuable insights into the performance and durability of $\text{Co}_{0.5}\text{Ni}_{0.5}@$ SDC catalysts, shedding light on the intricate interplay between catalyst composition, structure, and stability.

We have revised the following sentences in the revised manuscript to describe the above results:

“Post-catalytic characterizations revealed that while the elemental composition and crystalline phase of all catalysts did not exhibit noticeable changes after the stability tests (Supplementary Fig. 16 and Table 6), the morphologies of them displayed distinct trends (Supplementary Figs. 17-20). The average particle sizes of metals in $\text{Ni}@$ SDC and $\text{Co}_{0.5}\text{Ni}_{0.5}@$ SDC increased notably by 60% and 104% following stability tests, respectively, whereas the increases in other $\text{Co}_x\text{Ni}_{1-x}@$ SDC catalysts ($x = 0.2, 0.5, 0.75,$ and 1.0) featuring encapsulated structures were all below 15% (Supplementary Fig. 19). Meanwhile, the increase of the average particle sizes of SDC in all catalysts remained below 10% (Supplementary Fig. 20).” “During the long-term stability tests, $\text{Co}_{0.5}\text{Ni}_{0.5}@$ SDC displayed modest changes in both the EIS and DRT spectra, whereas other catalysts experienced higher increases of R_p and LF peaks (Supplementary Figs. 25-27), underscoring the durability of $\text{Co}_{0.5}\text{Ni}_{0.5}@$ SDC.”

Comment 3: Regarding the structure of Co-Ni alloy encapsulated in SDC, the authors have investigated it by SEM-EDX and TEM-EDX in the main text and supplementary information (Fig. 3d, Supplementary Figs. 6a and 7a). However, the reviewer think that relying solely on these data can only show that Co-Ni alloy and SDC indeed mixed together. These data cannot offer conclusive evidences that Co-Ni alloy encapsulated in SDC. Similarly, the Ni encapsulated in SDC cannot be confirmed solely by the characterization presented in the manuscript. Further characterizations, such as spherical aberration corrected transmission electron microscopy, are required.

Our response: As you suggested, we conducted spherical aberration corrected TEM characterization. The spherical aberration corrected EDX mapping clearly illustrates the encapsulated structure of $\text{Co}_{0.5}\text{Ni}_{0.5}@SDC$, wherein Co and Ni form a homogeneous core encapsulated by Sm, Ce, and O on the surface (Supplementary Fig. 11a). However, despite attempts with spherical aberration corrected high-angle annular dark-field scanning TEM (HAADF-STEM) to achieve atomic resolution, the interference from the outer SDC layer prevented clear visualization of the Co-Ni alloy. Instead, atomic resolution was only achieved at the edge of the SDC layer (Supplementary Fig. 11b). This limitation is attributed to the requirement of sample thickness below 30 nm for achieving atomic resolution, which was not feasible for our sample. Consequently, we opted not to pursue this technique for the $\text{Ni}@SDC$ sample.

To supplement the TEM analysis, we conducted XPS and ICP analyses to investigate surface and bulk elemental distributions. The XPS measurements revealed surface Co and/or Ni contents of ~15 mol% for $\text{Co}_{0.5}\text{Ni}_{0.5}@SDC$ and $\text{Ni}@SDC$, lower than their respective bulk contents of ~80 mol% as determined by ICP (Supplementary Fig. 12). In contrast, the surface Co and Ni content for $\text{Co}_{0.5}\text{Ni}_{0.5}-SDC$ was ~60 mol%, closely mirroring its bulk content of ~80 mol% (Supplementary Fig. 12). These new results further provide robust evidence supporting the encapsulated structure of the Co-Ni alloy by SDC in $\text{Co}_{0.5}\text{Ni}_{0.5}@SDC$, the encapsulated Ni structure by SDC in $\text{Ni}@SDC$, and the simple composite structure of Co-Ni alloy and SDC in $\text{Co}_{0.5}\text{Ni}_{0.5}-SDC$.

We have added the following sentences in the revised manuscript to describe the above results:

“X-ray photoelectron spectroscopy (XPS) measurements revealed surface Co and Ni contents of ~15 mol% for $\text{Co}_{0.5}\text{Ni}_{0.5}@SDC$ and $\text{Ni}@SDC$, lower than their bulk contents of ~80 mol% as determined by ICP (Supplementary Fig. 12). In contrast, the surface Co and Ni content for $\text{Co}_{0.5}\text{Ni}_{0.5}-SDC$ was ~60 mol%, close to its bulk content of ~80 mol% (Supplementary Fig. 12). This result further confirms the encapsulated structure in $\text{Co}_{0.5}\text{Ni}_{0.5}@SDC$ and $\text{Ni}@SDC$, and the composite structure in $\text{Co}_{0.5}\text{Ni}_{0.5}-SDC$.”

Comment 4: On page 8 lines 6-8 of the main text, the authors conclude that $\text{CoNi}@SDC$ has the effect of suppressing metal agglomeration by comparing the SEM-EDX and TEM-EDX of $\text{CoNi}@SDC$, $\text{CoNi}-SDC$ and $\text{Ni}@SDC$ before and after the test, but this proof is still not convincing enough. It is recommended to supplement the particle size distribution evolution of the three samples before and after the reaction to prove the effect of $\text{CoNi}@SDC$.

Our response: We agree with you. We supplemented the particle size distribution evolution as suggested. As shown in Supplementary Fig. 19, the average particle sizes of metals in $\text{Ni}@SDC$ and $\text{Co}_{0.5}\text{Ni}_{0.5}-SDC$ increased notably by 60% and 104% following stability tests, respectively, whereas the increase for $\text{Co}_{0.5}\text{Ni}_{0.5}@SDC$ was below 5%. Meanwhile, the increase of the average particle sizes of SDC in all three catalysts was lower than 10% (Supplementary Fig. 20). These new results further confirm that $\text{Co}_{0.5}\text{Ni}_{0.5}@SDC$ has the effect of suppressing metal agglomeration.

We have added the following sentences in the revised manuscript to describe the above results:

“The average particle sizes of metals in Ni@SDC and Co_{0.5}Ni_{0.5}-SDC increased notably by 60% and 104% following stability tests, respectively, whereas the increase in other Co_xNi_{1-x}@SDC catalysts (x = 0.2, 0.5, 0.75, and 1.0) featuring encapsulated structures were all below 15% (Supplementary Fig. 19). Meanwhile, the increase of the average particle sizes of SDC in all catalysts remained below 10% (Supplementary Fig. 20).”

Comment 5: Supplementary Fig. 11, the authors prove that CoNi@SDC has the effect of suppressing coke formation. However, Raman testing is the statistical analysis of surface signals in a certain area at the micro-scale. The results are potentially accidental because it is possible to select areas that do not have carbon deposition. Carbon sulfur analyzer maybe offer a more objective and accurate statistical method for carbon content detection. I suggest the author to supplement this data.

Our response: As suggested by you, we conducted carbon elemental analysis by a combustion method for the revised manuscript. Consistent with the Raman measurements, carbon elemental analysis revealed a carbon content of about 1.8 wt% in Ni@SDC after stability test, while no detectable carbon was observed in other catalysts (Supplementary Table 7). We have revised the following sentence to describe the result:

“Raman measurements and carbon elemental analysis revealed coke formation on Ni@SDC after stability tests,^{24,25} while no carbon was detected in other catalysts (Supplementary Fig. 21 and Table 7).”

Comment 6: In the CoNi@SDC catalyst, is the metal active site Co or Ni, or both? If the active site is a single metal, what role does the other metal play in the CoNi@SDC catalyst? The author should present rigorous experimental results to clarify this aspect.

Our response: Our investigation reveals that both Co and Ni are involved in this catalyst. To substantiate this finding, we have conducted additional experiments and rigorous analyses for the revised manuscript.

Firstly, we examined the Co and Ni molar ratios in the Co_xNi_{1-x}@SDC catalysts, as illustrated in Supplementary Fig. 3. We observed that the current density and ECSA-normalized current density exhibited an increase with rising Co content up to 0.5. However, a further increase in Co content resulted in a decline in current density (Supplementary Fig. 3). This observation suggests a synergistic effect between Co and Ni in Co_xNi_{1-x}@SDC, indicating the necessity for an optimal ratio.

Secondly, we investigated a physical mixture catalyst of Ni@SDC and Co@SDC with segregated Ni and Co phases, denoted as Ni@SDC+Co@SDC (Supplementary Fig. 7). This catalyst bears a similar composition to Co_{0.5}Ni_{0.5}@SDC featuring a Co-Ni alloy phase (Supplementary Table 2). Comparing the

current density and ECSA-normalized current density across different configurations, we observed the activity increased in the sequence of Co@SDC < Ni@SDC+Co@SDC < Ni@SDC < Co_{0.5}Ni_{0.5}@SDC (Supplementary Fig. 7). This observation provides further evidence of the synergistic effect between Co and Ni in Co_{0.5}Ni_{0.5}@SDC, which is not observed in Ni@SDC+Co@SDC.

In summary, our comprehensive analysis confirms that both Co and Ni are involved in Co_{0.5}Ni_{0.5}@SDC, contributing synergistically to its high CO₂ electroreduction activity.

We have added the following sentences in the revised manuscript to describe the above results:

“A correlation of the Co and Ni molar ratios to activity in the Co_xNi_{1-x}@SDC catalysts revealed that the current density increased with an increase of Co content up to x = 0.5, and a further increase in Co content decreased the current density (Supplementary Fig. 3).” “Additionally, a physical mixture of Ni@SDC and Co@SDC with segregated Ni and Co phases (denoted as Ni@SDC+Co@SDC) displayed inferior activity than Co_{0.5}Ni_{0.5}@SDC featuring a Co-Ni alloy phase (Supplementary Figs. 6 and 7).”

Comment 7: The CoNi@SDC catalysts showed no discernible morphological alterations after 2,000 h of electrolysis, whereas the Ni@SDC catalysts experienced significant structural changes after 500 h of electrolysis, exhibiting obvious metal agglomeration. Given that Ni metal encapsulated within the SDC underwent significant agglomeration, this strongly suggest that the encapsulation structure cannot prevent metal agglomeration. It is therefore that the Co-Ni alloy composition plays a significant role in inhibiting agglomeration and coking formation in the CoNi@SDC catalyst. How the alloy component with a Co: Ni ratio of 0.5:0.5 inhibits metal agglomeration and coking formation? How about other Co:Ni ratios?

Our response: To address your question, we have extended our investigation to encompass the stability analyses of additional Co_xNi_{1-x}@SDC catalysts, including compositions featuring x = 0.2, 0.75, and 1.0. Our expanded study reaffirms that while Ni and Co individually face distinct challenges—Ni experiencing coke formation and Co prone to severe oxidation and mass loss—an optimally balanced Co-Ni alloy composition can mitigate these issues. Indeed, the encapsulation structure emerges as a key factor in suppressing metal agglomeration.

The additional stability test results, shown in Supplementary Fig. 8, showcase notable trends. Co_{0.2}Ni_{0.8}@SDC demonstrated promising long-term stability, although inferior to the performance of Co_{0.5}Ni_{0.5}@SDC, with a degradation rate of 0.20 mV/h over 1,000 h of operation at 1 A/cm². In contrast, Co_{0.75}Ni_{0.25}@SDC and Co@SDC exhibited limited long-term stability, evidenced by significantly increased cell voltages and higher degradation rates.

Post-catalytic characterizations revealed that while the elemental composition and crystalline phase of all catalysts did not exhibit noticeable changes after the stability tests (Supplementary Fig. 16 and Table 6), their morphologies displayed distinct trends (Supplementary Figs. 17-20). The average particle

sizes of metals in Ni@SDC and Co_{0.5}Ni_{0.5}-SDC increased notably by 60% and 104% following stability tests, respectively, whereas the increase in other Co_xNi_{1-x}@SDC catalysts ($x = 0.2, 0.5, 0.75, \text{ and } 1.0$) featuring encapsulated structures were all below 15% (Supplementary Fig. 19). Meanwhile, the increase of the average particle sizes of SDC in all catalysts remained below 10% (Supplementary Fig. 20). Raman measurements and carbon elemental analysis revealed coke formation on Ni@SDC after stability tests, while no carbon was detected in other catalysts (Supplementary Fig. 21 and Table 7). Post-stability XPS and XAS spectra revealed severe Co oxidation in Co@SDC, contrasting with modest changes in the oxidation states of Co and Ni in Ni-containing catalysts (Supplementary Fig. 22 and 23), likely due to the higher oxygen affinity of Co relative to Ni (*J. Phys. Chem. C* **125**, 21902-21913 (2021)). Moreover, catalysts with higher Co content suffered from mass loss (Supplementary Table 8). These results underscore the pivotal role of encapsulated structure in mitigating metal agglomeration and effectiveness of optimal Co-Ni alloy composition in suppressing coke formation, metal oxidation and mass loss, contributing synergistically to enhanced stability. The encapsulated structure in Ni@SDC failed to inhibit Ni agglomeration, possibly due to coke formation on the Ni surface disrupting the encapsulated structure.

The new results have been added in the revised manuscript.

Comment 8: TPD measurements showed that the CoNi@SDC catalysts exhibited higher temperatures for CO₂ desorption compared to Ni@SDC at temperature ≥ 500 °C, while lower temperatures were observed for CO desorption in this temperature range. This suggests that the Co-Ni alloy enhanced the adsorption of CO₂ but weakening the adsorption of CO when compared to pure Ni in this temperature range. Thus, the authors attributed the higher activity of Co-Ni alloy to stronger CO₂ adsorption and weaker CO binding. However, the evaluation of CO₂ electroreduction performance is conducted at a temperature of 800 °C, which is significantly higher than the temperature range of the TPD measurements. The reviewer reasonably believe that the TPD results cannot accurately represent the strength of CO/CO₂ adsorption by the catalyst at the actual reaction temperature of 800 °C.

Our response: We appreciate your comment regarding the interpretation of our TPD results in the context of the actual reaction temperature. It is indeed crucial to consider the electric field effect, which TPD measurements inherently overlook. As noted in previous studies (*J. Am. Chem. Soc.* **139**, 11277–11287 (2017)), a negative electric field can enhance the adsorption of CO₂.

To address this limitation and provide a more accurate representation of CO₂/CO adsorption under reaction conditions, we firstly tried operando Raman measurements at 800 °C. However, we failed to detect any CO₂/CO species probably due to the low resolution and severe molecular vibration at high temperatures (Supplementary Fig. 15).

Then, we conducted DRT analyses based on operando EIS measurements at 800 °C. The analysis revealed distinct peaks corresponding to various electrochemical processes as discussed in Comment 1

(*Nat Energy* **7**, 866–875 (2022); *Nat. Commun.* **12**, 5665 (2021)). Specifically, the intensity of the low-frequency (LF) peaks reflects the adsorption ability, with lower intensities indicating higher adsorption and activation ability. Under CO₂ atmospheres, the LF peaks decreased in the order of Ni@SDC > Co_{0.5}Ni_{0.5}-SDC > Co_{0.5}Ni_{0.5}@SDC (Fig. 4c), indicating enhanced CO₂ adsorption and activation in the sequence of Ni@SDC < Co_{0.5}Ni_{0.5}-SDC < Co_{0.5}Ni_{0.5}@SDC. In contrast, under CO atmospheres, the LF peaks decreased in Co_{0.5}Ni_{0.5}-SDC > Co_{0.5}Ni_{0.5}@SDC > Ni@SDC (Fig. 4d), suggesting weaker CO activation for Co_{0.5}Ni_{0.5}-SDC and Co_{0.5}Ni_{0.5}@SDC compared to Ni@SDC. These additional analyses provide robust evidence supporting our initial conclusions regarding the enhanced CO₂ adsorption and activation capability of the Co-Ni alloy catalyst compared to pure Ni under the actual reaction conditions.

We have added the following sentences in the revised manuscript to describe these new results:

“Regarding DRT analysis, it is established that the high-frequency (HF), intermediate-frequency (IF), and low-frequency (LF) peaks correspond to oxygen-ion migration in the electrolyte, surface oxygen transfer in the air side, and electrochemical adsorption and activation processes in the fuel side, respectively.^{17,28} The LF peaks decreased in Ni@SDC > Co_{0.5}Ni_{0.5}-SDC > Co_{0.5}Ni_{0.5}@SDC under CO₂, and Co_{0.5}Ni_{0.5}-SDC > Co_{0.5}Ni_{0.5}@SDC > Ni@SDC under CO, while IF and HF peaks remained consistent (Figs. 4c and 4d). Collectively, our results indicate that Co-Ni alloy enhances CO₂ adsorption and activation but weakens CO activation compared to Ni, with more presence of metal-oxide interfaces facilitating the activation processes of both CO₂ and CO. Pure SDC exhibited weak activation of both CO₂ and CO (Supplementary Fig. 24), consistent with its limited CO₂ electroreduction activity (Fig. 2a).”

Referee #2.

General Comment.

The authors have developed an encapsulated Co-Ni alloy electrocatalyst which exhibited good performance towards CO₂RR to produce CO, especially excellent in activity and stability at high current of 1 A/cm². They explained the good electrocatalytic performance for CO₂ reduction were attributed to the enhanced CO₂ adsorption, weaken CO adsorption and suppressed metal agglomeration. The manuscript demonstrates a commendable level of originality by introducing a facile approach to synthesize high-efficient and durable CO₂ reduction catalyst. Moreover, the findings presented in this work hold promise for large-scale CO₂ conversion to CO, underscoring the manuscript's high level of significance. Overall, this manuscript demonstrates a solid foundation of research and addresses an important topic in the field. The methodology is well-structured and the data analysis is thorough. However, there are several areas that need attention before considering it for publication.

Our response: We thank you for the positive evaluation of our work, and for valuable suggestions. We have carried out new experiments and revised our work accordingly.

Specific Comments.

Comment 1: The normalized activity based on BET surface areas showed interesting results, however, for the electrocatalytic activity, the normalized activity based on electrochemically active surface area (ECSA) is more convincing than specific surface area. What are the ECSA normalized activities? Also, the BET surface area normalized activities of CoNi-SDC \approx CoNi@SDC, which means that these two catalysts have the same intrinsic activity? The author should further explain the role of encapsulation. Does the metal oxides actually block the active sites (CoNi) by encapsulation? Then the author actually did trade the activity with the stability.

Our response: Thank you for these comments, which prompted us to further enhance the clarify and depth of our manuscript. We have conducted ECSA measurements using a double-layer capacitance (C_{dl}) method, as determined via EIS measurements in a symmetrical cell configuration (Supplementary Fig. 5 and Table 3, and please see Methods section for comprehensive details). The ECSA-normalized current density data reveal a distinct sequence: SDC < Ni@SDC < Co_{0.5}Ni_{0.5}-SDC < Co_{0.5}Ni_{0.5}@SDC at all investigated cell voltages (Supplementary Fig. 4b). This result indicates that the Co_{0.5}Ni_{0.5}@SDC catalyst, with its encapsulated structure, has higher intrinsic activity compared to the Co_{0.5}Ni_{0.5}-SDC catalyst lacking such encapsulation.

We discuss and clarify the roles of encapsulation in the revised manuscript. Firstly, encapsulation effectively suppresses metal agglomeration, as extensively discussed in the response to Comment 2 below. Secondly, encapsulation contributes to an increased presence of metal-oxide interfaces, although it may reduce the exposure of metal sites, as discussed in the response to Comment 4 below. These

metal-oxide interfaces serve as the active sites for high-temperature CO₂ electroreduction, as oxides function as oxygen-ion transport channels. The combined effects of suppressed metal agglomeration and increased metal-oxide interfaces through encapsulation synergistically contribute to the enhanced stability and activity of Co_{0.5}Ni_{0.5}@SDC.

We have integrated these results and discussion in the revised manuscript.

Comment 2: In terms of the stability, I don't think the CoNi-SDC experienced a significant increase in cell voltage from 1.21 to 1.51 V after 500 h of operation, as the author stated. Within the first several hours, the voltage increased rapidly from 1.21 to 1.51 V, which is obviously a potential increase that is a normal phenomenon in the chronopotentiometry curve. The potential of CoNi-SDC stabilized at 1.51 within 500 h. To me, CoNi-SDC is stable and it is important to know what happens after 500 h to 2000 h. If the activity drops, we can conclude that encapsulation can inhibit the activity drop, otherwise such a conclusion is not supported by reliable data evidence.

Our response: We agree with your comment. In response to your suggestion, we extended the stability test duration from 500 h to 800 h in the revised manuscript. The updated results showed that although the cell voltage increase was more severe within the first 100 h (from 1.21 V to 1.45 V), it continued to increase to 1.51 V at 500 h and further to 1.60 V at 800 h under 1 A/cm² for Co_{0.5}Ni_{0.5}-SDC (Fig. 2d). In contrast, the cell voltage over Co_{0.5}Ni_{0.5}@SDC only marginally increased from 1.10 V to 1.13 V at 100 h and further to 1.20 V at 2,000 h (Fig. 2d). Therefore, we clarify that the observed rapid voltage increase within the initial hours for Co_{0.5}Ni_{0.5}-SDC cannot attributed to the chronopotentiometry method, as this method was also employed for Co_{0.5}Ni_{0.5}@SDC. Instead, it indicates a rapid catalyst deactivation process for Co_{0.5}Ni_{0.5}-SDC. These findings support our conclusion that encapsulation plays a crucial role in inhibiting activity drop over prolonged operation periods.

We have also revised the following sentences in the revised manuscript:

“In contrast, the Co_{0.5}Ni_{0.5}-SDC catalyst lacking an encapsulated structure, despite exhibiting high initial activity and ~100% FE_{CO}, experienced a significant increase in cell voltage from 1.21 to 1.60 V after 800 h of operation (Fig. 2d), with a degradation rate of 0.49 mV/h.”

Comment 3: The author concluded that the Co-Ni alloy enhanced the adsorption of CO₂ while weakening the adsorption of CO when compared to pure Ni in this temperature range, then what makes CoNi@SDC better than CoNi-SDC when is has similar CO₂ and CO adsorption/desorption ability? Is this reasonable? Because CO tends to adsorb on metal sites (Ni-Co) which are encapsulated by metal oxides. The observed decrease in CO adsorption in CoNi@SDC is more reasonable. So far, it is clear to me why CoNi@SDC and CoNi-SDC show better activity and stability than Ni@SDC, but it is not clear why CoNi@SDC > CoNi-SDC because the results show that CoNi-SDC with good stability (data until

500 h) and intrinsic activity similar to encapsulated CoNi@SDC.

Our response: In response to your comment, we clarify the differences observed between Co_{0.5}Ni_{0.5}@SDC and Co_{0.5}Ni_{0.5}-SDC.

Firstly, in terms of the intrinsic activity and stability, our data (including the new ones) confirm that Co_{0.5}Ni_{0.5}@SDC indeed exhibits higher intrinsic activity and stability compared to Co_{0.5}Ni_{0.5}-SDC (see also responses to your Comments 1 and 2 above). Secondly, we acknowledge the need to elucidate why Co_{0.5}Ni_{0.5}@SDC surpasses Co_{0.5}Ni_{0.5}-SDC despite their similar CO₂ and CO binding strengths. The underlying reason for this difference lies in the increased presence of metal-oxide interfaces facilitated by encapsulation in Co_{0.5}Ni_{0.5}@SDC. These metal-oxide interfaces serve as the active site for high-temperature CO₂ electroreduction, as oxides function as oxygen-ion transport channels.

We have integrated these differences in the revised manuscript:

“We further normalized the current densities based on electrochemical active surface areas (ECSA), and the normalized activity followed a consistent trend (Supplementary Figs. 4b, 5 and Table 3).”
“Collectively, our results indicate that Co-Ni alloy enhances CO₂ adsorption and activation but weakens CO activation compared to Ni. Moreover, the encapsulated catalyst benefits from a higher area of metal-oxide interfaces, which promotes the oxygen ion transport via the oxides.”

Comment 4: Are there interactions (electronic and structural) between CoNi and SDC and between Co and Ni? It is not clear how the presence of these two phases in the system. XPS and/or XAS are necessary to study the electronic structures of the catalysts.

Our response: In response to your comment, we have conducted additional analysis using XPS and XAS techniques to elucidate the electronic structures of the catalysts.

The results, as presented in Figs. 3g, 3h, Supplementary Figs. 13 and 14, shed light on the interactions between CoNi and SDC in our catalysts. The quasi in-situ XPS spectra of Ni 2p (Supplementary Fig. 13a) and Co 2p (Supplementary Fig. 13b) unveiled a reduction in the binding energy of metallic Ni in Ni@SDC and Co in Co@SDC compared to their pure forms, respectively. Meanwhile, the Ce 3d XPS spectra (Supplementary Fig. 13c) revealed a decreased Ce³⁺ content for both samples compared to pure SDC. These trends suggest a charge transfer from Ce³⁺ to either Ni or Co within each sample. This phenomenon persisted upon alloying, with the oxidation states of Co and Ni maintained below zero, and a slight increase in Ce³⁺ content observed compared to their monometallic counterparts. Such alterations are conducive to enhanced oxygen-ion transport. Moreover, Co_{0.5}Ni_{0.5}@SDC showed an enhanced charge transfer compared to Co_{0.5}Ni_{0.5}-SDC, suggesting an increased presence of metal-oxide interfaces through encapsulation. Consistent with XPS findings, XAS spectra corroborated the observed charge transfer from oxide to metal in all samples, with Co_{0.5}Ni_{0.5}@SDC demonstrating more metal-oxide interfaces than Co_{0.5}Ni_{0.5}-SDC (Figs. 3g and 3h,

Supplementary Fig. 14, Supplementary Tables 4 and 5).

It is difficult to detect the charge transfer between Co and Ni in our catalysts experimentally due to the strong interactions between CoNi and SDC. Instead, we used theoretical simulations to investigate their electronic interaction. As shown in Supplementary Table 9, Bader charge analysis indicated a charge transfer from Co to Ni upon alloying.

We have added the following paragraph to describe the newly added results:

“The quasi in-situ XPS spectra of Ni 2p (Supplementary Fig. 13a) and Co 2p (Supplementary Fig. 13b) unveiled a reduction in the binding energy of metallic Ni in Ni@SDC and Co in Co@SDC compared to their pure forms, respectively. Meanwhile, the Ce 3d XPS spectra (Supplementary Fig. 13c) revealed a decreased Ce³⁺ content for both samples compared to pure SDC. These trends suggest a charge transfer from Ce³⁺ to either Ni or Co within each sample. This phenomenon persisted upon alloying, with the oxidation states of Co and Ni maintained below zero, and a slight increase in Ce³⁺ content observed compared to their monometallic counterparts. Such alterations are conducive to enhanced oxygen-ion transport.⁴ Moreover, Co_{0.5}Ni_{0.5}@SDC showed an enhanced charge transfer between Co-Ni and SDC compared to Co_{0.5}Ni_{0.5}-SDC, suggesting an increased presence of metal-oxide interfaces through encapsulation. Consistent with XPS findings, X-ray absorption spectroscopy (XAS) and operando Raman spectra corroborated the observed charge transfer from oxide to metal in all samples, with Co_{0.5}Ni_{0.5}@SDC demonstrating more metal-oxide interfaces than Co_{0.5}Ni_{0.5}-SDC (Figs. 3g, 3h, Supplementary Figs. 14,15 and Tables 4,5).” “Bader charge analysis indicated a charge transfer from Co to Ni in all Co-Ni alloy models (Supplementary Table 9).”

Comment 5: For the statement in the introduction: “Because the reaction is conducted in a gas-phase process with pure CO₂ as reactant (without water), a complete selectivity for CO formation can be obtained.” It is better to explain why 100% CO selectivity in gas phase and cite reference if necessary.

Our response: In a low-temperature MEA electrolyzer, the presence of both gaseous CO₂ and liquid H₂O as reactants gives rise to the formation of various products, including CO, H₂ and hydrocarbons. In contrast, in the high-temperature SOEC, the exclusive use of pure gaseous CO₂ as the reactant, without H₂O or other hydrogen sources, results in the exclusive formation of CO without the generation of H₂ or other hydrogenation products (*Adv. Mater.* **31**, 1902033 (2019); *J. Electrochem. Soc.* **167**, 044508 (2020)). Consequently, this condition enables the achievement of complete selectivity for CO formation. We have revised the following sentences in the revised introduction to concisely describe the above explanation:

“This approach exclusively employs pure CO₂ as the reactant, without the inclusion of H₂O or other hydrogen sources, thereby affording complete selectivity for CO formation^{4,7}”

Comment 6: Figure 1b, you should reconsider naming the left block instead of “Current catalysts”. This is very general term. Do you mean that the current existing catalysts are all metal, metal+oxide, and oxide-supported metal? And encapsulated alloy in this work developed as a new term for the first time? Apparently that is not the case, see references: *ACS Catal.* 2023, 13, 8, 5135–5146 and *Applied Catalysis B: Environmental*, 283, 2021, 119628.

Our response: We agree with you and revised the description to “*Current CO₂ SOEC catalysts*”.

Comment 7: Since the title emphasized high-temperature CO₂ reduction, should the author provide the reaction temperatures for their performance data, for both ref. [4-8] and data from this study.

Our response: We agree with you and have added the reaction temperatures in the revised abstract:

“While high-temperature CO₂ electroreduction in a solid oxide electrolysis cell is industrially relevant, current catalysts suffer from a modest energy efficiency and a limited lifetime at high current densities, generally below 70% and 200 hours at 1 A/cm² and ≥800 °C, respectively.⁴⁻⁸ Here we develop an encapsulated Co-Ni alloy catalyst by Sm₂O₃-doped-CeO₂ that exhibits an energy efficiency of 90% and a lifetime of more than 2,000 hours at 1 A/cm² for high-temperature CO₂-to-CO conversion at 800 °C.”

Referee #3.

General Comment.

Review of “Encapsulated Co-Ni alloy boosts high-temperature CO₂ electroreduction”.

This is an interesting paper that proposes a new type of catalyst that boosts high temperature electroreduction. Overall, the paper is quite interesting and certainly has a significant amount of intellectual merit. The results also seem to be significant since the improved catalytic activity and the stability of the catalyst under consideration is quite significant. The data also seems to be of high quality for the presented work.

However, the referee does have two concerns associated with the paper that should be addressed before it can be accepted to such a prestigious journal:

Our response: We thank you for the positive evaluation of our work. We have revised the manuscript accordingly, see details below.

Specific Comments.

Comment 1: The results shown in Figure 2 are certainly interesting. However, in order to better understand the synergy between the two metals it would be also interesting to try other cases such as Co@SDC. Is there any particular reason why this cannot be done?

Our response: In response to your suggestion, we have extended our investigation to encompass the activity and stability analyses of additional Co_xNi_{1-x}@SDC catalysts, including compositions featuring x = 0, 0.2, 0.5, 0.75, and 1.0. Thus, Co@SDC is included here. Our updated findings uncover compelling trends in both activity and stability metrics.

The results are shown in Supplementary Figs. 3 and 8. Regarding activity, the current density and normalized current density based on ECSA increased with an increase in Co content up to 0.5, and a further increase in Co content rather decreased the current density (Supplementary Fig. 3). Similarly, the long-term stability tests demonstrated that the lifetime increased and degradation rate decreased with an increase in Co content up to 0.5 (Fig. 2d and Supplementary Fig. 8). A further increase in Co content resulted in a decrease in lifetime and an increase in degradation rate (Fig. 2d and Supplementary Fig. 8). Consequently, Co_{0.5}Ni_{0.5}@SDC exhibited the optimal combination of activity and stability. These trends indicate the presence of a synergistic effect between Ni and Co in our catalysts, wherein an optimal Co-Ni composition contributes to enhanced activity and stability for CO₂ electroreduction.

We have incorporated these additional findings into the revised manuscript to provide a more comprehensive understanding of the synergistic interactions between two metals in our catalysts:

“A correlation of the Co and Ni molar ratios to activity in the Co_xNi_{1-x}@SDC catalysts revealed that the

current density increased with an increase in Co content up to 0.5, and a further increase in Co content rather decreased the current density (Supplementary Fig. 3).” “The stability of other $\text{Co}_x\text{Ni}_{1-x}\text{@SDC}$ catalysts ($x = 0.2, 0.75, \text{ and } 1.0$) was also investigated, all showing inferior stability compared to $\text{Co}_{0.5}\text{Ni}_{0.5}\text{@SDC}$ (Supplementary Fig. 8).”

Comment 2: With regard to the results that are shown in Figure 3, the referee is unsure about the results that are shown and in particular what does the low temperature peak correspond to (in the 100 to 200 degree Celsius range). Some discussion in the text would be useful here.

Our response: Thank you for this comment. The low-temperature peak (below 200 °C) corresponds to the release of physisorbed CO_2 and CO (*Adv. Energy Mater.* **9**, 1803156 (2019); *J. Catal.* **359**, 8-16 (2018)). However, as discussed in the response to Comment 3 below, we have opted to remove the TPD results from our manuscript and used a more accurate DRT analyses to represent the CO_2/CO adsorption based on operando EIS measurements at 800 °C in the revised manuscript.

Comment 3: How should the reader understand the high temperature peak? This would correspond to a very unusually high binding energy of carbon dioxide. To have this paper published in such a prestigious publication as Nature, the referee thinks that there should be some sort of first principles-based calculations that would be reported in the manuscript. In this way, the reader would be better able to rationalize the results.

Our response: Thank you for this comment regarding the interpretation of the high-temperature CO_2 TPD peak, which has prompted us to reassess our approach to characterizing CO_2 adsorption. Upon careful consideration and consultation with established literature, including DFT simulations and experimental evidences, we have recognized that the high-temperature CO_2 TPD peak (500~800 °C) can be attributed to the decomposition of polydentate/tridentate/carbonate species on a reduced CeO_2 surface (*ACS Appl. Mater. Interfaces* **14**, 31862-31878 (2022)). Given that SDC (Sm_2O_3 -doped- CeO_2) is not the active site for CO_2 adsorption in our system, we have opted to remove the TPD results from our manuscript.

To address this limitation and provide a more accurate representation of CO_2/CO adsorption for our system, we have conducted DRT analyses based on operando EIS measurements at 800 °C in the revised manuscript. The analysis revealed distinct peaks corresponding to various electrochemical processes. Specifically, the intensity of the low-frequency (LF) peaks reflects the adsorption ability, with lower intensities indicating higher adsorption and activation ability (*Nat Energy* **7**, 866–875 (2022); *Nat. Commun.* **12**, 5665 (2021)). Under CO_2 atmospheres, the LF peaks decreased in the order of $\text{Ni@SDC} > \text{Co}_{0.5}\text{Ni}_{0.5}\text{-SDC} > \text{Co}_{0.5}\text{Ni}_{0.5}\text{@SDC}$ (Fig. 4c), indicating enhanced CO_2 adsorption and activation in the sequence of $\text{Ni@SDC} < \text{Co}_{0.5}\text{Ni}_{0.5}\text{-SDC} < \text{Co}_{0.5}\text{Ni}_{0.5}\text{@SDC}$. In contrast, under CO atmospheres, the LF

peaks decreased in $\text{Co}_{0.5}\text{Ni}_{0.5}\text{-SDC} > \text{Co}_{0.5}\text{Ni}_{0.5}\text{@SDC} > \text{Ni@SDC}$ (Fig. 4d), suggesting weaker CO adsorption and activation for $\text{Co}_{0.5}\text{Ni}_{0.5}\text{-SDC}$ and $\text{Co}_{0.5}\text{Ni}_{0.5}\text{@SDC}$ compared to Ni@SDC . These additional analyses further confirm the enhanced CO_2 adsorption and activation capability of the Co-Ni alloy catalyst compared to pure Ni under the actual reaction conditions.

Furthermore, DFT simulations were conducted to determine the adsorption energies of CO_2 and CO, as well as the energy profiles for CO_2 electroreduction to CO, on Co-Ni alloy and Ni models, as depicted in Supplementary Figs. 28-31. Slabs with (111) and (001) terminations were utilized, reflecting the predominant facets in our catalysts (Supplementary Fig. 6). In Comparison to Ni surfaces, Co-Ni alloy demonstrated either enhanced or comparable CO_2 adsorption, particularly evident on models featuring Co atoms on the outermost layer (Supplementary Fig. 29). In contrast, CO adsorption was weaker on Co-Ni alloy models across all facets compared to pure Ni (Supplementary Fig. 29). These trends persisted even when considering metal thermal expansions at 800 °C (Supplementary Fig. 30). Furthermore, the computed energy profiles for CO_2 electroreduction showed that Co-Ni alloy exhibited a lower reaction barrier compared to pure Ni (Supplementary Fig. 31). Consistent with DRT analysis, these DFT simulations indicate that Co-Ni alloy enhances CO_2 adsorption while weakens CO adsorption relative to Ni, thereby contributing to its superior CO_2 electroreduction to CO performance.

We have integrated these additional analyses and insights into the revised manuscript to provide a more comprehensive understanding of the underlying mechanisms governing CO_2 adsorption and electroreduction on our catalysts:

“Regarding DRT analysis, it is established that the high-frequency (HF), intermediate-frequency (IF), and low-frequency (LF) peaks correspond to oxygen-ion migration in the electrolyte, surface oxygen transfer in the air side, and electrochemical adsorption and activation processes in the fuel side, respectively.^{17,28} The LF peaks decreased in $\text{Ni@SDC} > \text{Co}_{0.5}\text{Ni}_{0.5}\text{-SDC} > \text{Co}_{0.5}\text{Ni}_{0.5}\text{@SDC}$ under CO_2 , and $\text{Co}_{0.5}\text{Ni}_{0.5}\text{-SDC} > \text{Co}_{0.5}\text{Ni}_{0.5}\text{@SDC} > \text{Ni@SDC}$ under CO, while IF and HF peaks remained consistent (Figs. 4c and 4d). Collectively, our results indicate that Co-Ni alloy enhances CO_2 adsorption and activation but weakens CO activation compared to Ni, with more presence of metal-oxide interfaces facilitating the activation processes of both CO_2 and CO. Pure SDC exhibited weak activation of both CO_2 and CO (Supplementary Fig. 24), consistent with its limited CO_2 electroreduction activity (Fig. 2a).” “Density functional theory (DFT-PBE-D3) simulations were conducted to determine the adsorption energies of CO_2 and CO, as well as the energy profiles for CO_2 electroreduction to CO, on Co-Ni alloy and Ni models, as depicted in Supplementary Figs. 28-31. Slabs with (111) and (001) terminations were utilized, reflecting the predominant facets in our catalysts (Supplementary Fig. 6). Bader charge analysis indicated a charge transfer from Co to Ni in all Co-Ni alloy models (Supplementary Table 9). In Comparison to Ni surfaces, Co-Ni alloy demonstrated either enhanced or comparable CO_2 adsorption, particularly evident on models featuring Co atoms on the outermost layer (Supplementary Fig. 29). In contrast, CO adsorption was weaker on Co-Ni alloy models across all facets compared to pure Ni (Supplementary Fig. 29). These trends persisted even when considering metal

thermal expansions at 800 °C (Supplementary Fig. 30).²⁹ Furthermore, the computed energy profiles for CO₂ electroreduction showed that Co-Ni alloy exhibited a lower reaction barrier compared to pure Ni (Supplementary Fig. 31). Consistent with DRT analysis, these DFT simulations indicate that Co-Ni alloy enhances CO₂ adsorption while weakens CO adsorption relative to Ni, thereby contributing to its superior CO₂ electroreduction to CO performance.”

Response to referees

We would like to thank the referees for reviewing our revised manuscript. They have provided additional comments and suggestions. We have revised the manuscript further according to these comments and suggestions. Below is a description of our point-by-point response. The changes in the main text and SI have been highlighted.

Referee #1.

General Comment.

The reviewer appreciates the responses from the authors that have addressed most of my concerns. The manuscript is greatly improved, but there are still some suggestions that the author may want to consider:

Our response: We thank you for the positive feedback on our work, and have revised the manuscript according to your suggestions.

Specific Comments.

Comment 1: According to previous research (*Angew. Chem. Int. Ed.* 2024, 63, e202313361), when CO₂ electrolysis is performed in SOEC, CO₂ molecules will first be adsorbed on the cathode surface to form a carbonate intermediate, and then the carbonate intermediate will accept electrons and dissociate into CO and O²⁻ at the three phase boundaries. Finally, CO is desorbed from the cathode surface, O²⁻ is transported to the anode through the electrolyte, and O₂ molecules are generated through the oxygen evolution reaction. Is it possible that the authors obtain a volcano-shaped curve about the CO₂ adsorption and activation and CO desorption of the different catalysts you tested? As such, this will provide more mechanistic guidance for audience to design catalysts.

Our response: Thank you for your insightful comment and for referencing the relevant research. In response to your suggestion, we have incorporated the relationship between CO₂ electroreduction activity (current density at 1.1 V) and the CO₂/CO adsorption and activation abilities (low-frequency peak intensity from DRT analysis) for the different catalysts tested in our revised manuscript. Our results indicate that CO₂ electroreduction activity is positively correlated with CO₂ adsorption and activation ability in our system (Fig. 4c). Additionally, we observed a volcano-shaped relationship between CO₂ electroreduction activity and CO adsorption and activation ability (Fig. 4d). These findings suggest that strong CO₂ adsorption and moderate CO adsorption are advantageous for CO₂ electroreduction in our system.

We have added the following sentence in the revised manuscript to describe the above results:

“A correlation between CO₂ electroreduction activity and CO₂/CO adsorption and activation

abilities for different catalysts showed that strong CO₂ adsorption and moderate CO adsorption are advantageous for CO₂ electroreduction in our system (Figs. 4c,d)."

Comment 2: It is recommended that the authors supplement the in-situ NAP-XAS test to detect the intrinsic reaction mechanism during carbon dioxide electrolysis. The difference in the production of key intermediates between different catalysts can be reflected by capturing the signals of carbonate species, thereby providing more comprehensive understandings for the catalytic performance differences.

Our response: We suspect that you may have intended to suggest "NAP-XPS" rather than "NAP-XAS". In response to your recommendation, we conducted in-situ near ambient pressure X-ray photoelectron spectroscopy (NAP-XPS) to probe the presence of carbonate species. It is reported that the peaks at ~289.3 eV in the C 1s spectrum and ~529.3 eV in the O 1s spectrum can be attributed to adsorbed carbonate species (*Angew. Chem. Int. Ed.* 2024, 63, e202313361). However, in our system, which utilized Sm₂O₃-doped CeO₂ in the catalyst, we found significant overlap between the Ce 4s and surface oxygen species signals with the carbonate C 1s and O 1s signals, respectively (Fig. R1). For example, in the case of the Co_{0.5}Ni_{0.5}@SDC and SDC catalysts, the high surface concentrations of Ce and oxygen species make it challenging to deconvolute the signals from adsorbed carbonates, which are present in much lower concentrations (Fig. R1). We also attempted operando Raman measurements but were unable to detect any carbonate species (Supplementary Fig. 14), probably due to the low resolution and severe molecular vibration at high temperatures.

Given these experimental limitations, we turned to DFT calculations to investigate the adsorption of carbonate species on different catalysts. Our DFT results showed that CO₂ adsorption was weak on isolated metal or SDC surfaces, whether or not oxygen vacancies were present in the SDC, with adsorption energies ranging from -0.05 to -1.00 eV (Supplementary Fig. 32). This suggests that neither the metal or SDC alone serves as an active site for CO₂ adsorption. Instead, the metal-SDC interface, represented here by Ce₃SmO₇ clusters adsorbed on metal surfaces, enhanced CO₂ adsorption as carbonates, with adsorption energies ranging from -1.83 to -2.08 eV (Supplementary Fig. 32). This indicates that the metal-SDC interface is the primary active site for CO₂ adsorption. Specifically, the adsorption energy of carbonate at the CoNi-SDC interface was calculated to be -2.08 eV, significantly lower than the -1.83eV observed at the Ni-SDC interface (Supplementary Fig. 32c), confirming enhanced CO₂ adsorption at the CoNi-SDC interface.

We have added the following sentences in the revised manuscript to describe the above results:

"CO₂ adsorption was weak on isolated metal or SDC surfaces, whether or not oxygen vacancies were present in the SDC, with adsorption energies ranging from -0.05 to -1.00 eV (Supplementary Fig. 32). This result suggests that neither the metal or SDC alone serves as an active site for CO₂ adsorption. Instead, the metal-SDC interface, represented here by Ce₃SmO₇ clusters adsorbed on metal surfaces, enhanced CO₂ adsorption as carbonates, with adsorption energies ranging from -1.83 to -2.08 eV

(Supplementary Fig. 32). This result indicates that the metal-SDC interface is the primary active site for CO_2 adsorption.” “Notably, CoNi-SDC demonstrated enhanced CO_2 adsorption and weakened CO adsorption compared to Ni-SDC (Supplementary Figs. 32c and 33c).”

Fig. R1 | In-situ NAP XPS measurements. **a**, C 1s spectra over $\text{Co}_{0.5}\text{Ni}_{0.5}@SDC$. **b**, O 1s spectra over $\text{Co}_{0.5}\text{Ni}_{0.5}@SDC$. **c**, C 1s spectra over SDC. **d**, O 1s spectra over SDC. The pressures of H_2 and CO_2 atmospheres were about 0.6 and 0.7 mbar, respectively.

Comment 3: Regarding the techno-economic analysis, the author mainly calculated the operating costs. Is it possible to provide additional calculations on how much economic benefits there would be if the device is used for commercial and stable operation to generate CO? This will help with evaluating the future development trend of high-temperature CO₂ electrolysis.

Our response: We would like to clarify that our preliminary cost estimation included both capital costs (electrolyzer, catalyst and electrolyte costs) and operating costs (CO₂, separation, heating, electricity, BoP and installation costs) (see Supplementary Note 1 for details).

Given that electricity, catalyst and electrolyte, and BoP and installation costs comprise the largest fraction of the total costs (Supplementary Table 17), we have conducted additional estimations to explore the economic benefits. These estimations consider the effects of electricity price, catalyst and electrolyte lifetime, and current density. Our results indicate that a reduction in electricity price leads to a linear decrease in CO production cost (Supplementary Fig. 37a). To achieve a profitable CO production based on the current market price (~450 \$/ton), the electricity price should be below 0.038 \$/kWh. This price point is attainable in recent onshore wind power auctions, such as those in Brazil, and Canada, where levelized electricity costs are as low as 0.03 \$/kWh (*Science* **364**, eaav3506 (2019)). Additionally, increasing the catalyst and electrolyte lifetime and current density (up to ~2 A/cm²) further reduces CO production costs (Supplementary Figs. 37b and 37c).

We have added the following sentence in the revised manuscript to describe the above results:

“A lower electricity price (below ~0.038 \$/kWh), a longer lifetime, and a higher current density (up to ~2 A/cm²) further contribute to the economic benefit of our system (Supplementary Fig. 37).”

Referee #2.

General Comment.

I have reviewed the revised manuscript and appreciate the substantial additional experiments conducted, including DFT calculations, in-situ XPS, XAS, EIS, and DRT analyses. These efforts significantly strengthen the manuscript by providing robust evidence of the excellent performance of the Co-Ni alloy catalyst encapsulated by Sm₂O₃-doped CeO₂ for the CO₂ to CO reaction.

However, I have some inquiries regarding the experimental analysis:

Our response: We thank you for the positive feedback on our revised work. We have carried out additional analysis and revised our work accordingly.

Specific Comments.

Comment 1: XAS Analysis: It appears that fitting analysis for the XAS data has not been performed. Fitting analysis is essential for quantitatively determining the oxidation state, coordination environment, and electronic structure of the catalytic materials. This information is crucial for understanding the active sites and configuration of the materials. Could the authors provide a specific reason for this omission?

Our response: As you suggested, we have now performed and discussed the fitting analysis of the XAS data. X-ray absorption near-edge structure (XANES) data revealed that the average oxidation states of Ni and Co in all fresh samples were below zero (Supplementary Table 4), suggesting a charge transfer from SDC to the Ni and Co sites. In addition, the average oxidation states of Ni and Co in $\text{Co}_{0.5}\text{Ni}_{0.5}@SDC$ were lower than those in $\text{Co}_{0.5}\text{Ni}_{0.5}\text{-SDC}$ (Supplementary Table 4), indicating an enhanced charge transfer from SDC to Co-Ni and thus an increased presence of metal-oxide interfaces through encapsulation. Extended X-ray absorption fine structure (EXAFS) spectra revealed similar bond lengths and slight variations in coordination numbers for Ni(Co)-Co(Ni) among the samples (Supplementary Tables 5 and 6), likely due to differences in particle sizes.

We have added the following sentences in the revised manuscript to describe the above analysis:

“Consistent with XPS findings, X-ray absorption near-edge structure (XANES) (Figs. 3g, 3h, and Supplementary Table 4) and operando Raman spectra (Supplementary Fig. 14) corroborated the observed charge transfer from oxide to metal in all samples, with $\text{Co}_{0.5}\text{Ni}_{0.5}@SDC$ demonstrating more metal-oxide interfaces than $\text{Co}_{0.5}\text{Ni}_{0.5}\text{-SDC}$. Extended X-ray absorption fine structure (EXAFS) spectra (Supplementary Fig. 15 and Tables 5,6) revealed similar bond lengths and slight variations in coordination numbers for Ni(Co)-Co(Ni) among the samples, likely due to differences in particle sizes.”

Comment 2: EIS Results: I noticed that the EIS results were presented, but no fitting analysis was conducted on these data. Fitting the EIS data could provide insightful information regarding the charge transfer resistance, double-layer capacitance, and other electrochemical parameters that are vital for interpreting the catalytic behavior in electrochemical systems. I suggest that the authors perform and discuss the fitting of the EIS data to better elucidate the electrochemical properties of the catalyst under study.

Our response: As you suggested, we performed and discussed the fitting of the EIS data. As shown in Supplementary Fig. 24, with increasing Co content in $\text{Co}_x\text{Ni}_{1-x}@SDC$, the charge-transfer resistances (R_p) of the fuel electrode under CO_2 first increased (up to $x = 0.5$) and then decreased, while they continuously decreased under CO. Compared to $\text{Co}_{0.5}\text{Ni}_{0.5}\text{-SDC}$, $\text{Co}_{0.5}\text{Ni}_{0.5}@SDC$ exhibited lower R_p for the fuel electrode under both CO_2 and CO. Meanwhile, all ohmic resistances (R_o), R_p of the air electrode, and double-layer capacitances remained similar. These results indicate that an optimal Co-Ni alloy enhances CO_2 adsorption and activation while moderating CO activation compared to monometallic Ni and Co. Additionally, compared to $\text{Co}_{0.5}\text{Ni}_{0.5}\text{-SDC}$, the encapsulated $\text{Co}_{0.5}\text{Ni}_{0.5}@SDC$ have more areas of metal-oxide interfaces, which promotes the CO_2 adsorption and oxygen ion transport and leads to

higher activity.

We have added the following sentences in the revised manuscript to describe the above results:

“The EIS results (Figs. 4a, 4b and Supplementary Fig. 24) showed that with increasing Co content in $\text{Co}_x\text{Ni}_{1-x}\text{@SDC}$, the polarization resistances (R_p) of the fuel electrode under CO_2 first increased (up to $x = 0.5$) and then decreased, while they continuously decreased under CO. Compared to $\text{Co}_{0.5}\text{Ni}_{0.5}\text{-SDC}$, $\text{Co}_{0.5}\text{Ni}_{0.5}\text{@SDC}$ exhibited lower R_p for the fuel electrode under both CO_2 and CO. Meanwhile, all ohmic resistances (R_o), R_p of the air electrode, and double-layer capacitances remained similar.”

Comment 3: In Table S12, the manuscript mentions a 2000-hour lifetime for the catalyst and membrane. Could the authors clarify how this figure was determined? Is it based on experimental data or extrapolated from shorter-term tests? And in Table S13, the cost of catalyst is as low as 21 \$/ton, did the author consider the complex synthesis procedure of core-shell structured catalysts (capital cost and operational cost) or only the cost of raw materials are taken into account.

Our response: The 2000-hour lifetime for the catalyst and electrolyte was determined based on our experimental data, as shown in Fig. 2d. Regarding the cost of catalyst and electrolyte, we re-estimated it based on the cost of all raw materials used, assuming a 100% atomic efficiency (see Supplementary Note 1 for details). Our estimation gave a catalyst and electrolyte cost of ~91 \$/ton CO for our system (Supplementary Table 17). While it is challenging to rigorously quantify the capital and operational costs for the synthesis procedure, we added 150% of the catalyst and electrolyte costs to account for these factors, based on reported models (*Nature*, **617**, 724–729 (2023); *Joule* **5**, 706-719 (2021)). The capital and operational costs for catalyst and electrolyte synthesis were included in the BoP and installation costs (Supplementary Note 1). Additionally, we would like to clarify that the unit of catalyst cost in our paper is “\$ per ton of CO”, rather than “\$ per ton of catalyst”.

We have integrated these results and discussions in the revised manuscript.

Additional General Comment.

By addressing these points, the manuscript could provide a more comprehensive understanding of the catalyst's behavior, contributing further to the field.

Our response: Thank you again for your valuable feedback. We appreciate your suggestions and have incorporated them to provide a more comprehensive understanding of the catalyst's behavior, further enhancing the contribution to the field.

Referee #3.

General Comment.

Reviewer 3 has re-reviewed the manuscript. The referee appreciated the additional information that was put into the manuscript to address comment 1. The referee does understand why the results for the TPD were removed.

The new DFT-based results are not compelling, however, for several reasons:

Our response: We thank you for the positive feedback on our revised work. We have carried out additional DFT calculations, provided more calculation details, and revised our manuscript accordingly.

Specific Comments.

Comment 1: The authors seem to assume a uniform distribution of Ni and Co in the lattice in the models that are considered. It is well-known that adsorbates can induce segregation effects where one of the metals that the adsorbate binds to will “pump” the metal atom to the surface. This seems to have not been considered by the authors and it should have been in order to have compelling results.

Our response: In response to your comment, we performed a detailed analysis of segregation energy (E_{seg}) and islanding energy (E_{isl}) across various Co-Ni alloy models, both with and without adsorbed CO. Our study considered 68 additional alloy configurations. As shown in Supplementary Fig. 31a, in the absence of adsorbed CO, the segregation of Ni atoms from the subsurface to the surface (replacing Co atoms) was slightly exothermic, with E_{seg} values from -0.11 to -0.26 eV. On the other hand, the segregation of Co atoms to the surface was slightly endothermic, ranging from 0.14 to 0.04 eV. Regarding the E_{isl} , all values were slightly exothermic, ranging from -0.02 to -0.07 eV (Supplementary Fig. 31b). Moreover, CO adsorption did not induce any significant segregation, as it had minimal impact on both E_{seg} and E_{isl} (Supplementary Fig. 31). Overall, the E_{seg} and E_{isl} values fall within narrow ranges of 0.40 and to 0.11 eV, respectively. Considering the role of configurational entropy, these results indicate no strong preference for a particular surface termination. Therefore, we can infer that assuming a uniform distribution of Ni and Co in the Co-Ni alloy lattice is a reasonable approximation.

We have added the following sentences in the revised manuscript to describe the above results:

“A uniform distribution of Ni and Co was assumed for Co-Ni alloy, as no strong preference for a particular surface termination was observed, even in the presence of adsorbed CO (Supplementary Fig. 31).”

Comment 2: More importantly, the adsorption energy of CO₂ is tiny (not more than -0.6 eV) and will not remain on the surface during the experiments at 800 degrees. It would seem to the referee that the

authors will need to consider the effect of the SDC on the adsorption energies. The referee would anticipate that the CO₂ would form a carbonate that would anchor the CO₂ to the surface. The referee believes that a more realistic model should be considered in the underlying reaction mechanism.

Our response: Thank you for your insightful comment. To address the impact of SDC on adsorption as carbonates and their corresponding energies, we developed new models representing SDC and its interface with metal surfaces.

Our results, as shown in Supplementary Fig. 32, showed that CO₂ adsorption was weak on isolated metal or SDC surfaces, whether or not oxygen vacancies were present in the SDC, with adsorption energies ranging from -0.05 to -1.00 eV (Supplementary Fig. 32). This suggests that neither the metal or SDC alone serves as an active site for CO₂ adsorption. Instead, the metal-SDC interface, represented here by Ce₃SmO₇ clusters adsorbed on metal surfaces, enhanced CO₂ adsorption as carbonates, with adsorption energies ranging from -1.83 to -2.08 eV (Supplementary Fig. 32). This indicates that the metal-SDC interface is the primary active site for CO₂ adsorption.

We have added the following sentences in the revised manuscript to describe the above results:

“CO₂ adsorption was weak on isolated metal or SDC surfaces, whether or not oxygen vacancies were present in the SDC, with adsorption energies ranging from -0.05 to -1.00 eV (Supplementary Fig. 32). This suggests that neither the metal or SDC alone serves as an active site for CO₂ adsorption. Instead, the metal-SDC interface, represented here by Ce₃SmO₇ clusters adsorbed on metal surfaces, enhanced CO₂ adsorption as carbonates, with adsorption energies ranging from -1.83 to -2.08 eV (Supplementary Fig. 32). This indicates that the metal-SDC interface is the primary active site for CO₂ adsorption.”

Comment 3: The authors mention that “Spin polarization was included when needed.” Could the authors be more specific on this? Since Co is magnetic, the referee would anticipate that all calculations involving Co should be spin polarized. If this is indeed the case, this should be mentioned explicitly.

Our response: We agree with you and have clarified this point in the revised methods:

“Spin polarization was included in all simulations involving Co, Ni and oxygen-defective SDC systems⁴⁵.”

Comment 4: The lattice parameters should be given for all the calculations. In particular, since Co is hexagonal closed packed, there should be more than one parameter when giving the lattice constants. This information is needed if the results are to be reproducible.

Our response: In response to your comment, we have added a new table in the supplementary information that includes all the optimized lattice parameters (see Supplementary Table 10 for details). In addition, we clarify that we employed a face-centered cubic (fcc) structure for all metals, based on

our experimental results (Supplementary Fig. 6). This choice aligns with literature findings, which indicate that the fcc structure is more stable at high temperatures than the hexagonal closed packed structure for the Ni- and Co-based metals (*J. Phys. Chem. Solids*, **137**, 109194 (2020); *Mater. Charact.* **93**, 79 (2014)).

Comment 5: The authors mention that the NEBs were calculated. The actual NEBs should be given in the supplementary information.

Our response: We agree with you and have added a new figure to the supplementary information, showing snapshots of the transition states and intermediates involved in CO₂ electroreduction to CO across all models in this study (see Supplementary Fig. 35 for details).

Referee #4.

General Comment.

The manuscript focuses on the synthesis of an encapsulated Co-Ni alloy catalyst doped with Sm₂O₃ and CeO₂ for high-temperature electrochemical CO₂ reduction. As noted, electrochemical CO₂ reduction is often limited by energy efficiency and the catalyst's limited lifetime at high temperatures. The superior catalytic performance of this structure is characterized and rationalized through both experimental and theoretical methods, presenting an efficient strategy to overcome the typical trade-off between activity and stability.

As a theorist, I concentrate my review on the computational aspects of the paper. Density Functional Theory (DFT) calculations are applied to determine the adsorption energies of CO₂ and CO, as well as the energy profiles for CO₂ electroreduction to CO. I have several important concerns about this section. The theoretical calculations section appears to be relatively undetailed, and more calculations need to be provided relative to the experimental work to provide a theoretical basis for the experimental findings. Specifically, the choice of computational methods and structures lacks proper justification, and the conclusions drawn from the calculations are not well-supported.

Our response: We thank you for the positive feedback on our revised work. We have carried out additional DFT calculations, provided more computational details, and revised our manuscript accordingly.

Specific Comments.

Comment 1: According to Fig. 3e, Co and Ni are randomly distributed; however, in the DFT-calculated structures, Co and Ni are arranged alternately. The authors should consider a wider range of Co and Ni

combinations in their calculation model. As a typical core-shell structure, the Sm₂O₃-doped-CeO₂ (SDC) likely plays a critical role in the stability of the Co-Ni alloy and may influence catalytic performance, yet it is entirely ignored in the current version. Considering the superior catalytic performance, it is suggested to conduct the DFT calculations based on a more realistic structural model.

Our response: Thank you for your valuable comment. We have taken your suggestions into account by exploring a broader range of Co and Ni configurations and the incorporating the effects of SDC to better represent a realistic structural model in our DFT calculations.

Firstly, regarding the distribution of Co and Ni, we performed a detailed analysis of segregation energy (E_{seg}) and islanding energy (E_{isl}) across various Co-Ni alloy models, both with and without adsorbed CO. Our study considered 68 additional alloy configurations. As shown in Supplementary Fig. 31a, in the absence of adsorbed CO, the segregation of Ni atoms from the subsurface to the surface (replacing Co atoms) was slightly exothermic, with E_{seg} values from -0.11 to -0.26 eV. On the other hand, the segregation of Co atoms to the surface was slightly endothermic, ranging from 0.14 to 0.04 eV. Regarding the E_{isl} , all values were slightly exothermic, ranging from -0.02 to -0.07 eV (Supplementary Fig. 31b). Moreover, CO adsorption did not induce any significant segregation, as it had minimal impact on both E_{seg} and E_{isl} (Supplementary Fig. 31). Overall, the E_{seg} and E_{isl} values fall within narrow ranges of 0.40 and to 0.11 eV, respectively. Considering the role of configurational entropy, these results indicate no strong preference for a particular surface termination. Therefore, we can infer that assuming a uniform distribution of Ni and Co in the Co-Ni alloy lattice is a reasonable approximation.

Secondly, to evaluate the influence of SDC on adsorption energies, we developed new models that simulate the SDC and its interface with metal surfaces (Supplementary Fig. 30). Our results, as shown in Supplementary Fig. 32, showed that CO₂ adsorption was weak on isolated metal or SDC surfaces, whether or not oxygen vacancies were present in the SDC, with adsorption energies ranging from -0.05 to -1.00 eV (Supplementary Fig. 32). This suggests that neither the metal or SDC alone serves as an active site for CO₂ adsorption. Instead, the metal-SDC interface, represented here by Ce₃SmO₇ clusters adsorbed on metal surfaces, enhanced CO₂ adsorption as carbonates, with adsorption energies ranging from -1.83 to -2.08 eV (Supplementary Fig. 32). This indicates that the metal-SDC interface is the primary active site for CO₂ adsorption. On the other hand, CO adsorption was more favorable on the metal sites (Supplementary Fig. 33). Notably, CoNi-SDC demonstrated enhanced CO₂ adsorption and weakened CO adsorption compared to Ni-SDC (Supplementary Figs. 32c and 33c). Based on these findings, we propose a dual-site reaction mechanism for our catalysts: CO₂ is captured as a carbonate at the metal-SDC interface, followed by the reduction of CO₂ to CO at the adjacent metal sites. The Gibbs free energy profiles for CO₂ electroreduction to CO further confirmed that the reaction was energetically favored on the dual sites, rather than on isolated metals or SDC, with CoNi-SDC being more favorable than Ni-SDC (Supplementary Figs. 35 and 36).

We have integrated these results and discussions in the revised manuscript.

Comment 2: The computational methods should be described explicitly to enable reproduction by other researchers. In the current version, spin polarization is not mentioned. Additionally, there is inconsistency in the plane-wave cutoff values: 450 eV is mentioned on line 5 of the DFT simulation section (Page 17), while 700 eV is noted on line 8, which is confusing. Furthermore, the validity of the computational methods is under question. For the Co-Ni system, the commonly used PBE functional may not correctly describe the correlation interactions. More accurate methods employing different functionals should be considered in this calculation.

Our response: Thank you for your comments, which prompted us to further clarify and enhance our manuscript.

Firstly, we have addressed the spin polarization in the revised method section: “*Spin polarization was included in all simulations involving Co, Ni and oxygen-defective SDC systems*⁴⁵.”

Secondly, regarding the inconsistency in plane-wave cutoff values, we have clarified this in the revised method section: “*A kinetic cut-off energy of 450 eV was used for most simulations, except for lattice parameter optimizations, where a higher cut-off of 700 eV was applied to avoid Pulay stress.*” Pulay stress refers to an artificial stress generated by the incompleteness of the plane-wave basis set when the cell volume changes. According to the VASP manual, a higher cutoff energy is recommended for simulations involving cell shape and volume relaxations, such as lattice parameter optimization (VASP, *Energy vs volume Volume relaxations and Pulay stress*, April 7, 2022; *Density Functional Theory: a Practical Introduction*, John Wiley & Sons, 2009).

Thirdly, as you suggested, we have evaluated CO adsorption on metal surfaces using two additional GGA functionals: RPBE and PBEsol (*Phys. Rev. B* **59**, 7413 (1999); *Phys. Rev. Lett.* **100**, 136406 (2008)). Our results showed that, compared to PBE, RPBE overestimates and PBEsol underestimates the adsorption energies (Supplementary Fig. 34a). However, the overall trends in adsorption energies across the different surfaces remain consistent across all three functionals. We have added the following sentence to describe the result: “*The choice of the functional, thermal expansion effect*³², and CO coverage did not affect the overall adsorption trend (Supplementary Fig. 34 and Table 12).”

Furthermore, to enhance reproduction, all DFT data can be found online in the ioChem-BD repository at <https://iochem-bd.iciq.es/browse/review-collection/100/68203/359dffab11176947069b7ce1> and <http://dx.doi.org/10.19061/iochem-bd-1-314>.

Comment 3: The coverage of molecules can significantly impact the calculated results. In electrochemical calculations, Gibbs free energy is commonly used rather than adsorption energy, as it considers vibrational frequency and entropy. Additionally, thermodynamic corrections are not included in the calculation of the CO₂ reduction profile (Fig. S31).

Our response: Firstly, we have now evaluated the effect of CO coverage on Ni and Co-Ni alloy surfaces by calculating the interaction energy (E_{int} , eV) between two CO molecules adsorbed at adjacent equivalent sites. Our results showed that E_{int} values were below 0.11 eV across all surfaces investigated, suggesting a weak repulsive interaction between CO adsorbates (Supplementary Table 12). Despite this repulsive interaction, the overall trend remains consistent: the Co-Ni alloy exhibits weakened CO adsorption compared to pure Ni (Supplementary Table 12).

Secondly, as suggested, we have calculated the vibrational contributions to enthalpy and entropy to determine the Gibbs free energies. We have also accounted for the rotational and translational contributions for the gas-phase CO₂ and CO molecules. The resulting Gibbs free energies for CO₂ and CO were displayed in Supplementary Figs. 32 and 33, respectively. The trends across different surfaces remain consistent, with CoNi-SDC showing enhanced CO₂ adsorption and weakened CO adsorption compared to Ni-SDC.

Furthermore, as you suggested, we have included the thermodynamic corrections in the calculation of the CO₂ reduction profile, see Supplementary Fig. 36 for details.

We have integrated these results and discussions in the revised manuscript.

Comment 4: As shown in Fig. S29, the adsorption of CO is much stronger than that of CO₂. What then is the driving force to displace the adsorbed CO molecules? Furthermore, the adsorption energies of CO₂ on CoNi(001) are -0.48 and -0.40 eV, while on Ni(001), it is -0.43 eV. These results do not convincingly illustrate enhanced CO₂ adsorption.

Our response: Thank you for this comment regarding the adsorption energies of CO₂ and CO, which has promoted us to reassess our DFT model for calculating CO₂ and CO adsorptions. Using a more realistic model that includes the metal-SDC interface, we found that the adsorption energies for CO₂ and CO are comparable, ranging from -1.83 to -2.08 eV for CO₂ and from -2.02 to -2.28 eV for CO (Supplementary Figs. 32 and 33; see also response to your Comment 1). In addition, the adsorption energy of CO₂ at the CoNi-SDC interface was calculated to be -2.08 eV, significantly lower than the -1.83 eV observed at the Ni-SDC interface (Supplementary Fig. 32c), confirming enhanced CO₂ adsorption. The driving force for CO desorption is primarily due to the higher entropy of the gas-phase molecule.

We have integrated these results and discussions in the revised manuscript.

Response to referees

We would like to thank the four referees for reviewing our revised manuscript. They are now satisfied with all experimental parts of the manuscript. Reviewers 3 and 4 have a few remaining comments and suggestions for the DFT computations. We have conducted further computations and revised our manuscript to address these comments and suggestions. Below is a description of our point-by-point response. The changes in the main text and SI have been highlighted.

Referee #1.

General Comment.

The authors have resolved all my concerns. I recommend the publication of this manuscript as it is.

Our response: We thank you for the positive feedback on our work.

Referee #2.

General Comment.

I have reviewed the revised manuscript and am satisfied with the changes made by the author. I agree that the manuscript is ready for publication in its current form.

Our response: We thank you for the positive feedback on our work.

Referee #3.

General Comment.

The referee appreciates that authors have done additional calculations for the new version of the paper. However, the referee still has concerns with the revised manuscript. As such, the theoretical calculations are still not that convincing.

Our response: Thank you for acknowledging our revision and for raising additional points that warrant our attention. We have carried out additional DFT calculations, provided more calculation details, and revised our manuscript accordingly.

Specific Comments.

Comment 1: The following points were adequately addressed by the authors:

1. Point 3 and 4 of referee 3 were adequately addressed by the authors.

Our response: Thank you for the positive feedback.

Comment 2: The following points should be addressed:

2. The referee does appreciate that the authors have done some additional calculations regarding the segregation energies, but there needs to be more details in order for the calculations to be reproducible since the configurations that the authors have used to calculate these values are not given in the associated document. All the associated structures should be given.

Our response: As suggested, we have added two new figures in the Supplementary Information that provide all the associated structures used for the segregation and islanding energy simulations (see Supplementary Figs. 33 and 34 for details). Moreover, all DFT data of this work can be found online in the ioChem-BD repository, at <https://iochem-bd.iciq.es/browse/review-collection/100/68203/359dffab11176947069b7ce1> (link for the reviewers) and upon publication at <http://dx.doi.org/10.19061/iochem-bd-1-314>. The atomic positions of all DFT models are provided in Supplementary Data. These additions ensure that the DFT simulations are fully reproducible.

Comment 3: Although the referee does appreciate the efforts that the referees have made on point 2. The referee still has some concerns regarding the revised calculations.

(i) It is somewhat unclear how the structures that are depicted in Supplementary Figure 32c were obtained. Indeed, it seems somewhat arbitrary how the structure was chosen. Also, it is mentioned in the caption that other structures were considered but only the most stable are shown. Those structures should be shown in the text.

(ii) The referee has two concerns regarding the Gibbs free energy calculations. First, the Gibbs free energies are computed at a given temperature at pressure. The values of the temperature and the pressure are not given in the SI. This essential detail should be given.

(iii) It is also unclear where the vacancies lie in Supplementary Figure 33b. This needs to be clearly shown in the figure.

(iv) It does not make any sense that an adsorption energy to the Gibbs free energies will change by 1.5 eV. There needs to be more details in the manuscript why we should have such a large change.

Our response: Thank you for the relevant comments. We have carefully addressed each of your concerns as follows.

(i) We have clarified the selection process for the depicted structures in the revised manuscript: *“The Ni-SDC and CoNi-SDC interfaces were represented by a Ce_3SmO_7 clusters adsorbed on $p(4 \times 4)$ slabs of Ni(001) and CoNi(001) surfaces, respectively. Six different Ce_3SmO_7 aggregates were generated*

based on $CeO_2(111)$ models, where a Ce atom was replaced by Sm and an oxygen vacancy was introduced. Various adsorption configurations of these clusters were then explored on the metallic surfaces, resulting in 18 and 24 different structures on Ni(001) and CoNi(001), respectively". Additionally, we have included a new figure in the Supplementary Information displaying snapshots of all assessed interface structures (Supplementary Fig. 31). Furthermore, we have added another figure presenting snapshots of the different CO₂ adsorption conformations explored on the metal-SDC models (Supplementary Fig. 36).

(ii) We have specified the temperature and pressure conditions in the revised manuscript: "*Gibbs free energies were computed at 800 °C and 1 atm of CO₂, in accordance with the experimental reaction conditions*".

(iii) Oxygen vacancies in the SDC models have now been clearly indicated with dotted white circles in all relevant figures.

(iv) The observed differences between potential energy (E_{ads}) and Gibbs free energy (G_{ads}) arise from significant entropic contributions at 800 °C, particularly in the adsorption and desorption processes. For gas-phase CO₂ and CO, translational and rotational contributions, obtained from Gaussian 09 at 800 °C, are as follows: $TS_{trans} = 1.73$ eV and $TS_{rot} = 0.61$ eV for CO₂; $TS_{trans} = 1.67$ eV and $TS_{rot} = 0.52$ eV for CO. For adsorbed CO₂ and CO, only vibrational entropy contributions were considered, following standard practices in heterogeneous catalysis (Chorkendorff, I.; Niemantsverdriet, J. W. *Concepts of modern catalysis and kinetics*, John Wiley & Sons, 2006). This explains the large difference between E_{ads} and G_{ads} . We have added the following sentences in the revised manuscript: "*The difference between E_{ads} and G_{ads} arises from significant entropic contributions at 800 °C, particularly in adsorption and desorption processes. For gas-phase CO₂ and CO, both translational and rotational contributions were included ($TS_{trans} = 1.73$ eV and $TS_{rot} = 0.61$ eV for CO₂; $TS_{trans} = 1.67$ eV and $TS_{rot} = 0.52$ eV for CO, obtained from Gaussian 09)¹¹, while only vibrational contributions were considered for adsorbed CO₂ and CO*". The calculation details for computing E_{ads} and G_{ads} were provided in the Methods section.

Comment 4: Regarding point 5. The referee has multiple concerns with what is presented in Figure 36. In the original review, the referee requested that the NEBs be shown. What is shown in Figure 36 is not the NEBs. There should be several images between the initial state, the transition state and the final state. Also, why are there no barriers that are computed in panel b and c of this figure? In addition, the Gibbs free energies are positive in panel a and b in the figure, which means that the process would not happen. It does not really make sense to show this. As for the process shown in panel c of this figure, the CO₂ is very weakly bound to the surface so that this process is quite unlikely, too.

Our response: Thank you for the relevant comments. We have carefully addressed each of your concerns as follows.

1. NEB Snapshots: We have added two new figures in the supplementary information, showing snapshots of the images used in the CI-NEB method to locate the transition states (Supplementary Figs. 41 and 44)

2. Reaction barriers: For the isolated SDC surface (previously panel b), the reaction profile is highly endergonic, with $\Delta G_{*CO_2 \rightarrow *CO+O*}$ ranging from 2.63 to 2.14 eV (Supplementary Fig. 42b). Given that the process is thermodynamically unfavourable (in line with the low CO₂ electroreduction activity observed for pure SDC in Fig. 2a), computing reaction barrier for isolated SDC surfaces is not meaningful. For the metal-SDC interface (previously panel c), we have now computed the reaction barriers and included them in the revised manuscript (Supplementary Figs. 43-45). We evaluated two different mechanisms: (i) CO₂ is captured at the interface, diffuses to nearby metallic surfaces, and undergoes reduction to CO; (ii) CO₂ is captured and reduced directly at the metal-SDC interface. Our results showed that the first mechanism was more favourable (Supplementary Fig. 45). In addition, the CoNi-SDC and Ni-SDC exhibit similar CO₂ dissociation barriers (Supplementary Fig. 45), suggesting that the superior CO₂ electroreduction performance of Co_{0.5}Ni_{0.5}@SDC primarily stems from its enhanced CO₂ adsorption and weakened CO adsorption compared to Ni@SDC.

3. Inclusion of data for completeness: While we agree that CO₂ electroreduction would not happen on the isolated metal and SDC surfaces (previously panels a and b), we have chosen to present these results for completeness and to provide the community with all relevant data (Supplementary Figs. 40-42). More importantly, these panels are included to show that although CO₂ reduction to CO is promoted by the metal surfaces, their CO₂ capture ability is limited on the metal-only systems. In contrast, SDC enhances CO₂ adsorption, but does not favor CO₂ reduction. Thus, these Gibbs energy profiles highlight the importance of the dual-site mechanism proposed in this work: CO₂ is captured in the interface between SDC and metals, while it is reduced in nearby metallic sites.

4. CO₂ binding strength at the metal-SDC interface: We respectfully disagree with the referee's interpretation regarding CO₂ binding strength at the metal-SDC interface. While potential energy values confirm strong bond formation, Gibbs free energy calculations indicate that this process is exergonic even at 800°C, with $\Delta G = -0.39$ eV for CoNi-SDC and -0.11 eV for Ni-SDC (Supplementary Fig. 45). Indeed, this mild adsorption is needed to limit the potential formation of recalcitrant carbonates that poison the catalysts.

We have integrated these results and discussions in the revised manuscript.

A General Comment: The authors claim that the active site is the one where we have an adsorption energy of about 2 eV. That could also mean that the CO₂ is too strongly adsorbed on the surface so that the surface would get poisoned with CO₂. This should be mentioned in the manuscript. As such, the discussion should be moderated.

Our response: We clarify that adsorption energies expressed in terms of potential energies ($E_{\text{ads,CO}_2}$) are more favourable than those in terms of Gibbs free energies ($G_{\text{ads,CO}_2}$), as shown in Supplementary Fig 35. For instance, while $E_{\text{ads,CO}_2}$ on the CoNi-SDC interface is -2.08 eV, the corresponding $G_{\text{ads,CO}_2}$ value is -0.39 eV, indicating mild adsorption. We have revised the manuscript to discuss adsorption energy based on Gibbs free energy rather than potential energy.

Referee #4.

General Comment.

The authors have included additional DFT calculations in the revised manuscript, detailing the calculations, structural motifs, and adsorption energy analysis. These enhancements have improved the quality of the work. However, there are a few suggestions for the DFT section:

Our response: Thank you for your positive feedback on our revised work. We have carefully considered your suggestions and revised our manuscript accordingly.

Specific Comments.

Comment 1: The decomposition of CO₂ on the surface generates oxygen, which can accumulate and passivate active sites. It's crucial to investigate how varying oxygen coverage affects reaction activity.

Our response: As you suggested, we have investigated the effect of oxygen coverage on CO₂ electroreduction on both CoNi-SDC and Ni-SDC surfaces (Supplementary Fig. 46). Our results showed that as oxygen coverage increased (from 0 to 6 oxygen atoms), the Gibbs free energy for $M^*CO_2 \rightarrow M^*CO + O^*$ also increased on both surfaces, in particular for Ni-SDC. This indicates that oxygen accumulation on the catalyst surfaces is detrimental to CO₂ electroreduction. However, it is important to note that under the applied voltage, the adsorbed oxygen primarily migrates to the anode for the oxygen evolution reaction (Fig. 1a). This is supported by the observation that the oxidation states of our catalysts remain almost unchanged before and after stability tests (Supplementary Figs. 13, 15, 22 and 23).

We have added these discussions in the caption of Supplementary Fig. 46 and revised the following sentence in the revised manuscript to incorporate them:

“The Gibbs free energy profiles for CO₂ electroreduction to CO further confirmed that the reaction was energetically favoured on the dual sites rather than on isolated metals or SDC, with similar reaction barriers on CoNi-SDC and Ni-SDC and small influence from oxygen coverage (Supplementary Figs. 40-46)”.

Comment 2: The NEB calculation is shown in SI Fig. 36. Activation barrier energy values should be included. Results suggest that the Ni(001) surface outperforms CoNi(001), with CoNi(001) performing comparably to itself. A detailed discussion of these NEB results is needed. Additionally, some calculation details are missing, such as thermal corrections for the NEB results and the calculated vibrational frequencies. The substrate may also influence NEB outcomes.

Our response: We appreciate your suggestions and have incorporated the requested details into the revised manuscript.

First, we have added the activation energies (E_a) values for CO₂ dissociation into CO and O on isolated metal and metal-SDC surfaces (Supplementary Figs. 42a and 45). However, we did not compute E_a on isolated SDC surfaces, as the reaction profile is highly endergonic, with $\Delta G^*_{\text{CO}_2 \rightarrow \text{CO} + \text{O}^*}$ ranging from 2.63 to 2.14 eV (Supplementary Fig. 42b).

Second, we clarify that isolated metal surfaces cannot effectively trap CO₂ (Supplementary Fig. 35a). Therefore, it is not meaningful to compare the reaction barriers of individual metals or alloys in this context, even though we have made all relevant data available for the community (Supplementary Fig. 42a). In contrast, for the metal-SDC surfaces, we have now included all reaction profiles, highlighting the crucial role of the oxide in trapping and activating CO₂ (Supplementary Figs. 43-45).

Third, we have discussed the NEB results in the revised manuscript: “CO₂ reduction to CO is promoted by metal surfaces, though their CO₂ capture ability is limited. The activation energies (E_a) obtained for the metal surfaces indicate that the (001) surfaces of Ni and CoNi generally outperform the (111) surfaces. Moreover, Ni(001) and the most favourable CoNi(001) model exhibit similar E_a , within the margin of DFT errors. In contrast, SDC enhances CO₂ adsorption, but does not favor CO₂ reduction, in agreement with the low CO₂ electroreduction activity observed for pure SDC (Fig. 2a)”.

Furthermore, we have included details on the thermal corrections applied to our NEB results: “Vibrational contributions to enthalpy and entropy for the Gibbs free energies of transition states were computed using standard approximations: the ideal gas model, rigid rotor, and harmonic oscillator. Only the reactants were considered for metals, while for oxidic systems, oxygen atoms were also included in the vibrational estimations due to their lower molecular weight”.

Comment 3: The adsorption of CO₂ and CO is affected by temperature and the reactants' partial pressure, which has been overlooked in the current calculations.

Our response: We have specified the temperature and pressure conditions in the revised manuscript: “Gibbs free energies were computed at 800 °C and 1 atm of CO₂, in accordance with the experimental reaction conditions”.

Comment 4: The authors should discuss the relationship between enhanced CO₂ adsorption and charge transfer, as this could provide valuable insights for material design.

Our response: As you suggested, we have computed the Bader charges of adsorbed CO₂ molecules on different metal-SDC interfaces in the revised manuscript (Supplementary Fig. 37). Our analysis showed that the total Bader charge of the adsorbed CO₂ molecule ranged from -0.20 to -0.30 |e⁻|, compared to 0.03 |e⁻| for gas-phase CO₂, indicating a charge transfer from the catalyst surface to the CO₂ molecule. However, no strong correlation was observed between the CO₂ adsorption energy and the degree of charge transfer (Supplementary Fig. 37). This suggests that the enhanced CO₂ adsorption cannot be attributed solely to charge transfer. Other factors, such as the basicity of the oxygen site where the molecule is adsorbed and the acidity of the metal atoms interacting with the CO₂ molecule, also influence the interaction.

We have added the following sentence in the revised manuscript to reflect these results: “*showing no strong correlation with the degree of charge transfer (Supplementary Fig. 37)*”.

Response to referees

We would like to thank Referee #4 for reviewing our revised manuscript. Referee #4 are now satisfied with our manuscript, so we make no further scientific revision.

Referee #4.

General Comment.

The manuscript has been substantially enhanced through the inclusion of additional calculations and discussions. It effectively demonstrates that both alloy composition and encapsulated structure are pivotal in determining the activity and stability of catalysts in high-temperature electrochemical CO₂-to-CO reactions. Upon thorough review of the responses to previous comments, the structural models now appear reasonable and reproducible, aligning well with the experimental results. The observed variations in activity across different structural models are particularly noteworthy, suggesting that CO₂ and CO may adsorb on two distinct sites rather than a single site. Although the synergistic interactions are not fully explored in the current approach, this version of the manuscript is thought-provoking and merits publication as is, as it paves the way for further research in this area.

Our response: We thank you for the positive feedback on our work.